# When Drafts Evolve: Speculative Decoding Meets Online Learning

**Yu-Yang Qian** [1 2]  **Hao-Cong Wu** [1 2]  **Yichao Fu** [3]  **Hao Zhang** [3]  **Peng Zhao** [1 2]

## Abstract

Speculative decoding has emerged as a widely adopted paradigm for accelerating large language model inference, where a lightweight draft model rapidly generates candidate tokens that are then verified in parallel by a larger target model. However, due to limited model capacity, drafts often struggle to approximate the target distribution, resulting in shorter acceptance lengths and diminished speedup. A key yet under-explored observation is that speculative decoding inherently provides *verification feedback* that quantifies the deviation between the draft and target models at no additional cost. This process naturally forms an iterative "draft commits–feedback provides–draft adapts" evolving loop, which precisely matches the *online learning* paradigm. Motivated by this connection, we propose OnlineSPEC, a unified framework that systematically leverages interactive feedback to continuously evolve draft models. Grounded in *dynamic regret minimization*, we establish a formal link between online learning performance and speculative system's acceleration rate, and develop novel algorithms via modern online learning techniques, including optimistic online learning that adaptively reuses historical gradients as predictive update hints, and online ensemble learning that dynamically maintains multiple draft models. Our algorithms are equipped with theoretical justifications and improved acceleration rates, achieving up to 24% speedup over seven benchmarks and five foundation models.

## 1. Introduction

Large language models (LLMs) have achieved remarkable success across a wide range of tasks (Brown et al., 2020; Radford et al., 2021; Ouyang et al., 2022). Recent advances in test-time scaling further extend the inference length for enhanced capabilities, enabling techniques such as chain-of-thought (Wei et al., 2022), LLM reasoning (Guo et al., 2025), and agent systems (Xi et al., 2025). However, this trend also increases the inference burden due to *sequential dependency* inherent in autoregressive models: each token can only be generated after its predecessor has been produced.

To reduce the inference latency of LLMs, *speculative decoding* (Leviathan et al., 2023; Chen et al., 2023) has emerged as a widely adopted paradigm. It can be viewed as a representative instance of the broader *generation-refinement framework* (Hu et al., 2025), in which a lightweight draft model generates a draft sequence that is then verified by a larger target model. Related approaches in this framework include cascade decoding (Varshney & Baral, 2022) and multi-token prediction (Cai et al., 2024). Most existing methods typically focus on *offline training* to obtain a strong draft model and keep it fixed during deployment. However, due to the capacity gap between the draft and target models, a fixed draft model cannot fully capture all knowledge domains of the target. As a result, it may fail to approximate the target distribution for diverse user inputs, leading to shorter acceptance lengths and degraded speedup.

To address this limitation, a critical observation is that the verification process inherently provides *interactive feedback*: each time the target model verifies a draft, it reveals precisely where the draft diverges from the target distribution. By further leveraging this feedback to refine the draft model, it naturally forms a "draft commits–feedback provides–draft adapts" evolving loop, enabling the draft model to continuously adapt and improve. Recent advances (Wang et al., 2025; Bhansali & Heck, 2025) start to explore such feedback. For example, OSD (Liu et al., 2024b) periodically updates the draft model at inference time using gradient descent based on the draft's error tokens. However, existing approaches mainly focus on the basic form of *token-level* feedback and employ ad-hoc algorithms designed for specific tasks or models, making it challenging to apply in broader scenarios. Overall, there still lacks a *principled way* to exploit interactive feedback in an online manner.

In this work, we propose a unified framework to systematically exploit the interactive feedback. Our key insight is that

---

[1]State Key Laboratory for Novel Software Technology, Nanjing University [2]School of Artificial Intelligence, Nanjing University [3]University of California, San Diego. Correspondence to: Peng Zhao <zhaop@lamda.nju.edu.cn>.

*Proceedings of the 43rd International Conference on Machine Learning*, Seoul, South Korea. PMLR 306, 2026. Copyright 2026 by the author(s).

**Figure 1.** Illustration of *Generation-refinement framework*. A draft sequence is first generated rapidly by a small draft model, and then verified by the target model, which naturally forms an iterative "draft commits–feedback provides–draft adapts" evolving loop.

the generation-refinement framework can be formulated as *online learning* (Hazan, 2016), a well-established problem where a player iteratively makes decisions and observes feedback from the environment. Building on this observation, we propose OnlineSPEC (*Speculative Decoding via Online Learning*), a unified framework to formulate draft-target interaction grounded in online learning: at each round, the draft model (player) generates a draft sequence, the target model (environment) verifies it and provides feedback, and the draft model updates accordingly. More importantly, for the first time, we establish a theoretical connection between the speculative system's *acceleration rate* and the online algorithm's *dynamic regret* (Zhang et al., 2018), i.e., the performance gap against time-varying comparators.

Our OnlineSPEC framework enables principled algorithm design by leveraging the rich toolkit from online learning, offering both systematic methodology and theoretical justifications. We demonstrate its generality through three instantiations that integrate with existing methods: *(i) Online-LR*, which applies online gradient descent with DPO-style loss for LR (Fu et al., 2025) in reasoning tasks; *(ii) Opt-Hydra*, which incorporates optimistic online learning (Rakhlin & Sridharan, 2013) into Hydra (Ankner et al., 2024) by reusing historical gradients as predictive hints; and *(iii) Ens-Eagle*, which employs online ensemble learning (Zhao et al., 2024) to adaptively combine multiple draft heads from EAGLE (Li et al., 2024b; 2025) for robust performance improvements. Experiments across seven benchmarks and five foundation models demonstrate the effectiveness of OnlineSPEC: our methods consistently outperform both offline baselines and naive online adaptations, achieving up to 24% speedup over previous SOTA methods while maintaining output quality.

**Organization.** Section 2 presents OnlineSPEC framework and theoretical foundations. Section 3 introduces its instantiations through newly proposed algorithms. Experiments are reported in Section 4, and finally we conclude in Section 5.

## 2. A Unified View from Online Learning

In this section, we first introduce the problem formulation, then propose a unified view grounded in online learning.

### 2.1. Problem Formulation

Speculative decoding and related acceleration methods, including cascade decoding and multi-token prediction, can be unified under the *generation-refinement framework* (Hu et al., 2025): a lightweight draft model first generates a candidate sequence, which is then verified or refined by a larger target model. These methods share a common challenge: due to limited capacity, the draft model often fails to fully approximate the target distribution, leading to degraded acceptance rate and speedup.

Importantly, a key observation is that the *interactive feedback* is available during the generation-refinement process, i.e., the verification process inherently reveals where drafts diverge from the target. If we leverage this feedback to evolve the draft models, which naturally forms a "draft commits–feedback provides–draft adapts" evolving loop. This iterative process precisely matches the online learning paradigm (Hazan, 2016), an iterative game between a *player* and an *environment*: at each round, the player commits to a decision, observes feedback from the environment, and uses this feedback to update its strategy. This connection allows us to use established tools from online learning to systematically understand and improve generation-refinement methods. Based on this insight, as detailed in Algorithm 1, we define OnlineSPEC framework as the following two stages:

- *Offline Initialization*: Get good initial draft model(s) $\mathbf{w}_0 \in \mathcal{W}$, which aims to predict the output distribution of the target model $\mathbf{v} \in \mathcal{V}$. Target model's parameter space is much larger than draft model's, i.e., $|\mathcal{V}| \gg |\mathcal{W}|$.

- *Online Adaptation*: At each step, the draft model produces a candidate sequence for verification by the target model. Tokens are accepted based on the likelihood ratio between the target and draft distributions. The draft model is then updated utilizing this verification feedback.

The key observation is that, the *acceptance rate*, i.e., the expected probability of accepting a draft token,

$$\text{Acc}_t \triangleq \mathbb{E}_{x \sim q_{\mathbf{w}_t}} \left[ \min \left\{ 1, \frac{p_{\mathbf{v}}(x \mid \mathbf{x})}{q_{\mathbf{w}_t}(x \mid \mathbf{x})} \right\} \right], \qquad (1)$$

directly determines the accepted length and speedup ratio.

Therefore, the goal is to *continuously evolve the draft model* to improve the acceptance rate during deployment.

## 2.2. Formulated as an Online Learning Problem

We now formulate the *online adaptation stage* as an online learning problem. Let $T$ denote the total number of generation steps, $k$ the candidate (draft) length, and $A$, $a$ the expected inference time of the target and draft model, respectively. At each round $t$, the draft model $\mathbf{w}_t$ generates a candidate sequence, which is verified by the target model $\mathbf{v}$. The draft model then receives feedback in the form of a loss function $f_t(\mathbf{w}_t)$ and updates itself to $\mathbf{w}_{t+1}$. We adopt *dynamic regret* (Zhang et al., 2018) as the performance measure, defined as the cumulative gap between the algorithm and a sequence of time-varying comparators $\{\mathbf{w}_t^\star\}_{t=1}^T$:

$$\mathbf{Reg}_T \triangleq \sum_{t=1}^T f_t(\mathbf{w}_t) - \sum_{t=1}^T f_t(\mathbf{w}_t^\star), \qquad (2)$$

where $f_t$ is cross-entropy loss obtained from target model $\mathbf{v}$.

**Remark 1** (Why Dynamic Regret?). We adopt *dynamic regret* rather than the standard *static regret* (Cesa-Bianchi & Lugosi, 2006) due to the inherent capacity gap between the draft and target models. Specifically, the draft model has significantly lower capacity and thus cannot globally match the target distribution $p_\mathbf{v}(\cdot \mid \mathbf{x})$ across all possible contexts $\mathbf{x}$. However, for any *specific* context sampled from step $t$, it is more reasonable to assume the existence of a local optimum $\mathbf{w}_t^\star$ such that the draft distribution aligns with the target on this particular context. This motivates the use of dynamic regret, which compares against a time-varying sequence $\{\mathbf{w}_t^\star\}_{t=1}^T$ of locally optimal draft models, rather than requiring a single fixed comparator across all steps.

One of our contributions is establishing a *formal connection* between regret minimization and the acceleration rate. Below, we derive this relationship, adopting assumptions following Leviathan et al. (2023) to facilitate analysis.

**Assumption 1.** Following Leviathan et al. (2023), we assume that the conditional distributions $q_{\mathbf{w}_t}(x \mid \mathbf{x}_{<i})$ are *i.i.d.* across all positions $i \in \{1, \dots, k\}$ at step $t$, and similarly for $p_\mathbf{v}(x \mid \mathbf{x}_{<i})$. We adopt the cross-entropy loss, i.e., $f_t(\mathbf{w}) = -\mathbb{E}_{x \sim p_\mathbf{v}(\cdot \mid \mathbf{x})}[\log q_\mathbf{w}(x \mid \mathbf{x})]$.

Under this assumption, we establish the following lemma regarding the expected accepted length.

**Lemma 1** (Accepted Length). *Under Assumption 1, for Algorithm 1 with $T$ steps, the expected length of the draft sequence $\mathbb{E}[|\hat{\mathbf{x}}|] = \sum_{t=1}^T \mathbb{E}[n_t]$ satisfies*

$$\frac{(k+1)\,T}{1 + (k+1)\sqrt{\mathbf{Reg}_T/(2T)}} \le \mathbb{E}[|\hat{\mathbf{x}}|] \le (k+1) \cdot T.$$

We are now ready to derive the relationship between the acceleration rate $\gamma$ and the dynamic regret $\mathbf{Reg}_T$.

---

**Algorithm 1** OnlineSPEC Framework

**Input:** Initial draft model parameter $\mathbf{w}_0 \in \mathcal{W}$ with its predicted distribution $q_{\mathbf{w}_0}(\cdot \mid \mathbf{x})$, target model $\mathbf{v} \in \mathcal{V}$ with its predicted distribution $p_\mathbf{v}(\cdot \mid \mathbf{x})$, total steps $T$, candidate length $k$, input prompt $\mathbf{x}_0$.

**Initialize:** Input sequence $\mathbf{x} = \mathbf{x}_0$, draft model $\mathbf{w}_t = \mathbf{w}_0$.

**for** $t = 1, \dots, T$ **do**
  ▷ Generate the draft sequence:
  **for** $i = 1$ **to** $k$ **do**
    $x_i \sim q_{\mathbf{w}_t}(x \mid \mathbf{x}_{<i})$, where $\mathbf{x}_{<i} = \{\mathbf{x}, x_1, \dots, x_{i-1}\}$.
  ▷ Verify by target model:
  Compute $\{p_\mathbf{v}(x_1|\mathbf{x}_{<1}), \dots, p_\mathbf{v}(x_k|\mathbf{x}_{<k})\}$ in parallel.
  ▷ Determine the accepted token length $n_t$:
  Sample $r_1 \sim U(0,1), \dots, r_k \sim U(0,1)$ uniformly;
  $n_t \leftarrow \min(\{j-1 \mid 1 \le j \le k, r_j > \frac{p_\mathbf{v}(x_j|\mathbf{x}_{<j})}{q_{\mathbf{w}_t}(x_j|\mathbf{x}_{<j})}\} \cup \{k\})$.
  ▷ Verify and update the draft sequence:
  **if** $n_t < k$ **then**
    $p'(x) \propto \max\left(0, p_\mathbf{v}(x \mid \mathbf{x}_{<n_t+1}) - q_{\mathbf{w}_t}(x \mid \mathbf{x}_{<n_t+1})\right)$;
  **else**
    $p'(x) \leftarrow p_\mathbf{v}(x \mid \mathbf{x}_{<k+1})$.
  $x_{n_t+1} \sim p'(x)$;  $\mathbf{x} \leftarrow \mathbf{x} + [x_1, \dots, x_{n_t}, x_{n_t+1}]$.
  ▷ Receive interactive feedback: observe the loss function $f_t : \mathcal{W} \mapsto \mathbb{R}$ from the target model $\mathbf{v}$.
  ▷ Update draft model: $\mathbf{w}_{t+1} \leftarrow \text{Update}(f_t; \mathbf{w}_t)$ as Sec. 3.

**Output:** Token sequence $\hat{\mathbf{x}} \triangleq [x_1, x_2, \dots]$.

---

**Theorem 1** (Acceleration Rate). *Under Assumption 1, for our OnlineSPEC with a total of $T$ steps, candidate length $k$, and inference times $A$ and $a$ for the target and draft models, respectively, the acceleration rate $\gamma$ satisfies*

$$\frac{k+1}{(\alpha k + 1)\left(1 + (k+1)\sqrt{\mathbf{Reg}_T/(2T)}\right)} \le \gamma \le \frac{k+1}{\alpha k + 1},$$

*where $\alpha \triangleq \frac{a}{A} \ll 1$ is the ratio of the inference times between the draft model and the target model.*

Theorem 1 demonstrates that the acceleration rate $\gamma$ depends on three factors: the algorithm's regret, the candidate length $k$, and the inference time ratio $\alpha$: a high acceleration rate can be achieved by using a *fast and accurate* draft model and selecting an appropriate candidate length $k$. It also highlights the importance of incorporating interactive feedback to *continuously evolve* the draft model, as this reduces the regret over time. We defer the proof to Appendix D.

**Remark 2** (Special Cases of Theorem 1). Two special cases of Theorem 1 are of interest. *(i)* When $\mathbf{Reg}_T = 0$, meaning all draft tokens are accepted, the acceleration rate achieves its maximum value $\gamma \approx k + 1$. *(ii)* In contrast, when $\mathbf{Reg}_T = \Theta(T)$, the lower bound reduces to $\Theta(1)$, indicating that no significant speedup is guaranteed.

**Remark 3** (Importance of Exploiting Interactive Feedback). If interactive feedback is not employed in the generation-

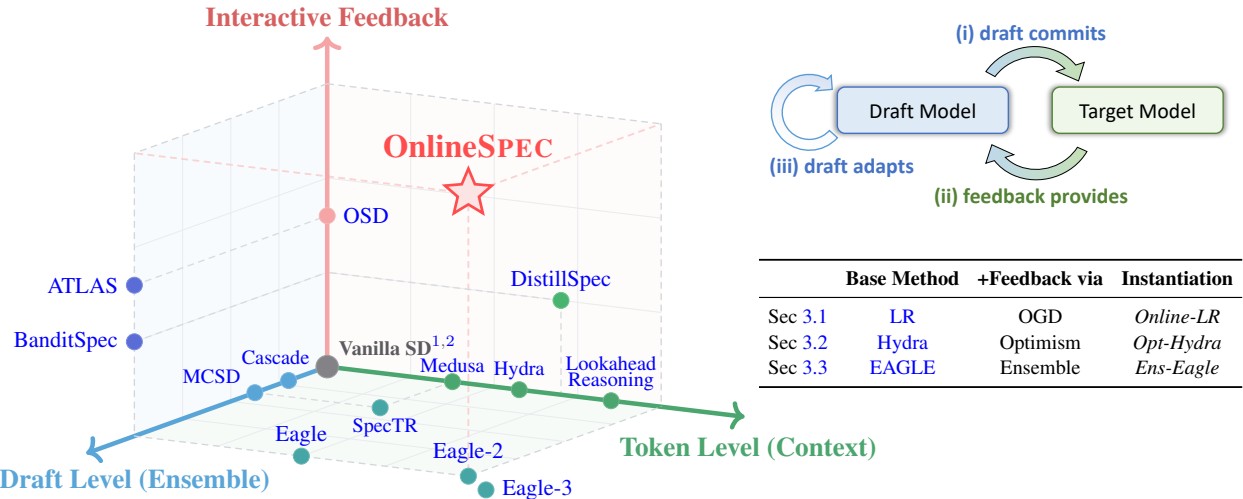

**Figure 2.** A comprehensive 3D-visualization illustrates generation-refinement approaches across three dimensions: draft level, token level, and incorporating interactive feedback. Our OnlineSPEC framework provides a unified perspective for integrating interactive feedback and can be *seamlessly* combined with existing methods to further enhance the acceleration rate.

refinement framework, which will cause $\mathbf{Reg}_T/T$ to be a constant throughout the inference process, then the denominator $1 + (k+1)\sqrt{\mathbf{Reg}_T/(2T)}$ grows with $k$, and the acceleration rate $\gamma$ converges to a constant. In contrast, if interactive feedback is exploited and $\mathbf{Reg}_T/T$ is optimized to be *sublinear*, the denominator diminishes as $T$ grows and the acceleration rate $\gamma$ approaches its maximum $\frac{k+1}{\alpha k+1}$, highlighting the necessity of leveraging interactive feedback to continuously evolve the draft model.

## 3. OnlineSPEC with Instantiations

In this section, we demonstrate how our OnlineSPEC framework can guide the design of new algorithms. As summarized in Figure 2, prior work has focused primarily on three aspects: *(i)* the *draft level*, which aims to improve the accepted length by ensembling multiple draft models or draft sequences; *(ii)* the *token level*, which aims to improve token-level prediction by employing more powerful models or incorporating additional contextual information; and *(iii)* the *interactive feedback* perspective, which aims to exploit verification feedback from the target model.

OnlineSPEC provides a unified perspective that systematically integrates interactive feedback into existing speculative methods. By leveraging the rich toolkit from online learning, it enables principled algorithm design. We demonstrate its generality through the following three instantiations:

1. *Online-LR*: Applying *online gradient descent* with DPO-style loss to Lookahead Reasoning (Fu et al., 2025).

2. *Opt-Hydra*: Incorporating *optimistic learning* in Hydra (Ankner et al., 2024) by historical gradients as hints.

3. *Ens-Eagle*: Employing *online ensemble learning* to combine multiple draft heads from EAGLE (Li et al., 2024b).

### 3.1. Online Update with Gradient Descent

We start with a classic algorithm in online learning, *online gradient descent* (OGD) (Zinkevich, 2003), which updates the model by moving alongside the negative gradient of the loss function. Specifically, at time step $t$, the update form is $\mathbf{w}_{t+1} = \Pi_{\mathcal{W}}\left[\mathbf{w}_t - \eta \nabla f_t(\mathbf{w}_t)\right]$, where $\Pi_{\mathcal{W}}[\mathbf{w}] = \arg\min_{\mathbf{w}' \in \mathcal{W}} \|\mathbf{w}' - \mathbf{w}\|_2$ denotes the projection operator, $\nabla f_t(\mathbf{w}_t)$ is the gradient at time $t$, and $\eta$ is learning rate.

This simple update rule is effective for adapting the draft model with interactive feedback and has been explored by the OSD (Liu et al., 2024b). Specifically, OSD tracks the token positions where the draft model generates incorrect predictions along with the corresponding target logits, and periodically updates the draft model by minimizing a distillation loss on these errors using the target logits as supervision. Although achieving notable success, OSD is specially designed for token-level error feedback in speculative decoding scenarios, and may fail to generalize to broader cases, for example, reasoning tasks, as shown in Table 2.

In contrast to task-specific designs, a key advantage of OnlineSPEC is its flexibility. While theoretical justifications are established under cross-entropy loss, the framework itself is flexible to the choice of loss function—by appropriately specifying $f_t(\cdot)$, it naturally extends to diverse scenarios, e.g., reasoning tasks (Fu et al., 2025) with a DPO-style loss (Rafailov et al., 2023), where feedback takes the form of preference pairs rather than token-level errors:

$$f_t(\mathbf{w}_t) = -\sum_{(x, y_w, y_l) \sim \mathcal{S}_t} \log \sigma\left(\beta \mathcal{L}(y_w) - \beta \mathcal{L}(y_l)\right),$$

where $(x, y_w, y_l)$ denotes a tuple of prompt $x$, preferred response $y_w$, and dispreferred response $y_l$ obtained from

the feedback set $\mathcal{S}_t$. Here $\sigma(\cdot)$ is the sigmoid function, $\mathcal{L}(y) \triangleq \log \frac{\pi_{\mathbf{w}_t}(y|x)}{\pi_{\text{ref}}(y|x)}$ is the log-likelihood ratio between the draft policy $\pi_{\mathbf{w}_t}$ and the reference policy $\pi_{\text{ref}}$, and $\beta$ controls the deviation from the reference policy. This instantiation yields the *Online Lookahead Reasoning* (Online-LR) algorithm, demonstrating how the unified framework accommodates different feedback structures. To facilitate the regret analysis, we introduce the following standard assumption in online convex optimization (Hazan, 2016):

**Assumption 2.** The domain $\mathcal{W}$ is bounded by $\|\mathbf{w}\|_2 \le D$, and the gradient is bounded by $\|\nabla f_t(\mathbf{w})\|_2 \le G, \forall \mathbf{w} \in \mathcal{W}$.

Under this assumption, we provide theoretical justification for the OGD-based approach as follows.

**Corollary 1.** Under Assumptions 1 and 2, when employing OGD with a learning rate $\eta = \mathcal{O}(1/\sqrt{T})$, the regret satisfies $\mathbf{Reg}_T \le \mathcal{O}(\sqrt{T}(1 + P_T))$, where $P_T \triangleq \sum_{t=1}^{T} \|\mathbf{w}_{t+1}^\star - \mathbf{w}_t^\star\|_2$ is the path length, and the acceleration rate $\gamma$ satisfies

$$\gamma = \Omega\left(\frac{1}{\alpha k + 1} \min\left\{k + 1, \frac{T^{1/4}}{\sqrt{1 + P_T}}\right\}\right).$$

Assumption 2 is commonly adopted in the online learning literature. While such convexity-based conditions may not strictly hold for neural network optimization, the resulting analysis still provides valuable justifications for the key factors governing the acceleration rate. Specifically, Corollary 1 demonstrates that OGD achieves good performance (sublinear regret) when the path length $P_T$ is small, i.e., when the change of the environment is smooth during deployment. In the context of the generation-refinement framework, this allows the draft model to gradually improve its prediction accuracy through timely feedback from the target model, thereby increasing the acceptance rate and overall efficiency in relatively smooth environments.

**Remark 4** (Comparison with OSD (Liu et al., 2024b)). OSD provides an initial exploration of leveraging interactive feedback in speculative decoding. However, OSD is specifically designed for token-level error feedback, which limits its applicability when the feedback structure differs. As shown in Table 2, directly adapting OSD to more diverse tasks such as reasoning, where feedback takes the form of preference pairs rather than token-level errors, leads to degradation in speed. In contrast, our OnlineSPEC framework offers a unified perspective grounded in online learning. By appropriately specifying the loss function and update scheme, our framework naturally extends to diverse feedback structures and enables systematic algorithm designs with theoretical justifications.

### 3.2. Online Update with Optimism

The previous OGD method evolves the draft model using only the current verification feedback. Can we further im-

prove performance by reusing historical gradient information to predictively adapt to the environment? This leads us to *optimistic online learning* (Rakhlin & Sridharan, 2013), a well-established technique in the online learning literature that has drawn considerable attention in the online learning community, especially for challenging problems (Wei & Luo, 2018; Zhao et al., 2024). This technique introduces predictive *hints* to guide model updates proactively. The motivation is that if we can accurately predict the update direction, we can adapt the draft model more effectively. Specifically, optimistic online learning performs the following *two-step* update procedure:

$$\mathbf{w}_t = \Pi_{\mathcal{W}}[\widehat{\mathbf{w}}_t - \eta \mathbf{h}_t]; \quad \widehat{\mathbf{w}}_{t+1} = \Pi_{\mathcal{W}}[\widehat{\mathbf{w}}_t - \eta \nabla f_t(\mathbf{w}_t)], \quad (3)$$

where $\nabla f_t(\mathbf{w}_t)$ is ground-truth gradient from verification feedback, $\widehat{\mathbf{w}}_t$ is an intermediate model, and $\mathbf{h}_t \in \mathcal{W}$ is a *hint* (or optimism) that serves as a guess of upcoming gradient.

It remains to construct the hint $\mathbf{h}_t$. In principle, hints can be constructed in various ways, especially if we have certain prior on evolving patterns of the gradients. A simple yet effective choice is to use the *last-round gradient* $\nabla f_{t-1}(\mathbf{w}_{t-1})$ as the hint. The intuition is that nearby user queries often exhibit temporal locality and similarity, so the gradient from the previous round serves as a reasonable approximation of the current one. Based on this design, we instantiate *Opt-Hydra*, which augments the *Hydra* (Ankner et al., 2024) using the last round's gradient as a hint. We provide the corresponding theoretical justification as follows.

**Corollary 2.** Under Assumptions 1 and 2, employing optimistic online learning in Eq. (3), if the hint approximates the true gradient with $\sum_{t=1}^{T} \|\mathbf{h}_t - \nabla f_t(\mathbf{w}_t)\|_2^2 \le \delta_T$, regret satisfies $\mathbf{Reg}_T \le \mathcal{O}(\sqrt{1 + \delta_T} \cdot (1 + P_T))$, and $\gamma$ satisfies

$$\gamma = \Omega\left(\frac{1}{\alpha k + 1} \min\left\{k + 1, \frac{\sqrt{T}}{(1 + \delta_T)^{1/4}\sqrt{1 + P_T}}\right\}\right).$$

Corollary 2 demonstrates that optimistic online learning can substantially improve performance when accurate hints are available. In the best case, when $\delta_T = \mathcal{O}(1)$, i.e., hints are sufficiently accurate on average, the dynamic regret becomes $\mathcal{O}(1 + P_T)$, improving upon standard OGD. This highlights importance of exploiting historical information to further enhance the performance of the draft model.

### 3.3. Online Update with Ensemble

The OGD-based methods discussed above achieve good performance when the environment is relatively stable. However, in open-world deployment scenarios (Zhou, 2022), user inputs may span diverse domains and shift over time. In this case, a fundamental limitation of OGD is that its performance depends heavily on the learning rate: a small

learning rate adapts slowly but stably, while a large one reacts quickly but may overshoot. Since the optimal learning rate depends on the unknown shifts of environment, a single learning rate cannot perform well across all scenarios.

To address this limitation, we draw inspiration from *online ensemble* paradigm (Zhao et al., 2024), which maintains a pool of *base learners* to handle different environments, and employs a *meta learner* to adaptively combine their outputs:

*Construct base learners with multiple step sizes.* We maintain $N$ base learners (draft models) with a set of learning rates $\mathcal{H} = \{\eta_i\}_{i=1}^N$. At round $t$, each base learner $\mathbf{w}_t^i$ is updated independently via OGD: $\mathbf{w}_{t+1}^i = \Pi_{\mathcal{W}}\left[\mathbf{w}_t^i - \eta_i \nabla f_t(\mathbf{w}_t^i)\right]$, where $\nabla f_t(\mathbf{w}_t^i)$ is the gradient of the loss function for the $i$-th base model. This yields a pool of different draft models $\{\mathbf{w}_t^i\}_{i=1}^N$.

*Combine the outputs by meta learner.* We employ a meta learner that combines the outputs of base learners through weighted averaging to obtain the final drafts: $\mathbf{w}_t = \sum_{i=1}^N p_t^i \cdot \mathbf{w}_t^i$, where weights $p_t^i \in [0,1]$ satisfy $\sum_{i=1}^N p_t^i = 1$. The weights are updated following exponential weighting scheme $p_t^i \propto \exp(-\varepsilon \sum_{s=1}^{t-1} f_t(\mathbf{w}_s^i))$, where $\varepsilon > 0$ controls the sensitivity to performance. Intuitively, the meta learner assigns higher weights to the base with smaller cumulative loss, thereby adaptively tracking the optimal one. The motivation is that different base learners excel at handling different environments, and the meta learner can adaptively track the best-performing one.

Building on this paradigm, we instantiate *Ens-Eagle* and *Ens-Eagle-3* by applying online ensemble to *EAGLE* (Li et al., 2024b) and *EAGLE-3* (Li et al., 2025), respectively. Specifically, we maintain multiple draft heads with different learning rates as base learners, and use Hedge algorithm (Freund & Schapire, 1997) as the meta learner to combine outputs. We now provide theoretical justification:

**Corollary 3.** Under Assumptions 1 and 2, employing online ensemble of a total of $N = \mathcal{O}(\log T)$ draft models with geometrically spaced learning rates, regret satisfies $\mathbf{Reg}_T \leq \mathcal{O}(\sqrt{T(1 + P_T)})$, and $\gamma$ satisfies

$$\gamma = \Omega\left(\frac{1}{\alpha k + 1} \min\left\{k + 1, \frac{T^{1/4}}{(1 + P_T)^{1/4}}\right\}\right).$$

Corollary 3 demonstrates that the online ensemble achieves a regret bound of $\mathcal{O}(\sqrt{T(1 + P_T)})$, which improves upon previous OGD's $\mathcal{O}(\sqrt{T}(1 + P_T))$ especially when the path length is large, i.e., when the domains of user input changes dramatically. This leads to a better acceleration rate, particularly in non-stationary environments where user inputs span diverse domains and shift over time. The ensemble approach adapts to such complicated environments by maintaining multiple drafts with different adaptation rates, and the meta learner adaptively tracks the best one on the fly.

# 4. Experiments

This section provides the experimental results of our approach. To comprehensively evaluate the effectiveness and efficiency of our OnlineSPEC framework and proposed approaches, we conduct experiments across seven benchmark datasets and three target models.[1] Our experimental evaluation aims to answer the following three research questions:

**Q1:** Do offline methods suffer from speedup degradation during deployment? Is interactive feedback necessary?

**Q2:** Can our OnlineSPEC be integrated with previous SOTA methods to achieve better speedup?

**Q3:** How do the hyperparameter choices affect the performance of the methods?

## 4.1. Experimental Setup

We first introduce the experimental setup as follows, including the contenders, implementation details, and datasets.

**Datasets.** We conduct experiments on seven benchmark datasets, including three math reasoning tasks *GMS8K* (Cobbe et al., 2021), *MATH* (Lewkowycz et al., 2022), and mathematically related subset of *MMLU* (Hendrycks et al., 2021); three code generation tasks *Code-Search-Python* (Husain et al., 2020), *Spider* (Yu et al., 2019), and *MBPP* (Austin et al., 2021); and a financial question answering dataset *Alpaca-finance* (Taori et al., 2023). These datasets span diverse domains and problem types and are widely used in the research community. We evaluate the performance of different methods using the following metrics: *(i)* performance metric (solve rate / pass@1 accuracy depending on the benchmark), *(ii) average accepted length*, i.e., how many tokens of the draft sequence are accepted in one speculative round by the target model on average, to measure the effectiveness of the draft models. *(iii) TPS* (tokens per second), which measures the wall-clock efficiency of the methods.

**Contenders.** We compare our proposed approach with the following state-of-the-art contenders: *(i)* vanilla speculative decoding (Leviathan et al., 2023; Chen et al., 2023). *(ii) OSD* (Liu et al., 2024b) which periodically updates the draft model using observed feedback via knowledge distillation. *(iii) Hydra* (Ankner et al., 2024) which improves draft head speculation by introducing sequential dependency among draft tokens. *(iv) EAGLE* (Li et al., 2024b) and *EAGLE-3* (Li et al., 2025) which predicts second-to-top-layer features using a lightweight draft head and creates a tree-structured draft sequence. *(v) LR* (*Lookahead Reasoning*) (Fu et al., 2025) which exploits step-level parallelism by proposing multiple future reasoning steps and verifying their semantic

---

[1]Our code is available at GitHub: https://github.com/ZinYY/OnlineSPEC

**Table 1.** Comparison of generation-refinement acceleration methods across different benchmark datasets. For each method, we report the *average accepted length* (AVGLEN ↑) and *wall-clock speedup ratio* (SPEEDUP ↑). Results are averaged over three runs with standard deviations. The best results are highlighted in bold.

| | GSM8K | | Spider | | Code-Search | | Alpaca-Finance | |
|---|---|---|---|---|---|---|---|---|
| | AVGLEN ↑ | SPEEDUP ↑ | AVGLEN ↑ | SPEEDUP ↑ | AVGLEN ↑ | SPEEDUP ↑ | AVGLEN ↑ | SPEEDUP ↑ |
| | | | | Target model: *lmsys / Vicuna-7B-v1.3* | | | | |
| Vanilla SD | 1.25±0.01 | 1.00× | 1.20±0.03 | 1.00× | 1.22±0.02 | 1.00× | 1.20±0.01 | 1.00× |
| **OSD** | **1.52±0.03** | **1.23×** | **1.36±0.02** | **1.12×** | **1.37±0.02** | **1.09×** | **1.30±0.01** | **1.08×** |
| Hydra | 2.14±0.05 | 1.00× | 2.65±0.31 | 1.00× | 1.82±0.04 | 1.00× | 1.78±0.01 | 1.00× |
| OSD-Hydra | 2.56±0.06 | 1.19× | 3.11±0.31 | 1.11× | 2.11±0.05 | 1.16× | 2.19±0.03 | 1.25× |
| (OnlineSPEC) **Opt-Hydra** | **2.69±0.08** | **1.26×** | **3.27±0.32** | **1.18×** | **2.37±0.07** | **1.31×** | **2.70±0.03** | **1.55×** |
| EAGLE | 1.48±0.03 | 1.00× | 1.31±0.03 | 1.00× | 1.41±0.03 | 1.00× | 1.39±0.01 | 1.00× |
| OSD-EAGLE | 1.92±0.04 | 1.28× | 1.53±0.03 | 1.21× | 1.54±0.04 | 1.07× | 1.51±0.02 | 1.09× |
| (OnlineSPEC) **Ens-EAGLE** | **2.01±0.05** | **1.41×** | **1.60±0.03** | **1.32×** | **1.58±0.04** | **1.15×** | **1.61±0.04** | **1.14×** |
| EAGLE-3 | 1.78±0.03 | 1.00× | 1.85±0.07 | 1.00× | 1.67±0.04 | 1.00× | 1.62±0.01 | 1.00× |
| OSD-EAGLE-3 | 2.07±0.04 | 1.20× | 2.10±0.05 | 1.07× | 1.97±0.04 | 1.13× | 2.00±0.04 | 1.16× |
| (OnlineSPEC) **Ens-EAGLE-3** | **2.16±0.03** | **1.26×** | **2.24±0.07** | **1.12×** | **2.01±0.07** | **1.18×** | **2.07±0.04** | **1.27×** |
| | | | | Target model: *meta-llama / Llama-2-7B-Chat* | | | | |
| Vanilla SD | 1.25±0.01 | 1.00× | 1.07±0.01 | 1.00× | 1.09±0.01 | 1.00× | 1.20±0.01 | 1.00× |
| **OSD** | **1.40±0.03** | **1.06×** | **1.47±0.05** | **1.22×** | **1.34±0.02** | **1.26×** | **1.31±0.01** | **1.12×** |
| Hydra | 1.52±0.02 | 1.00× | 1.48±0.08 | 1.00× | 1.66±0.01 | 1.00× | 1.64±0.01 | 1.00× |
| OSD-Hydra | 2.55±0.03 | 1.67× | 3.10±0.18 | 1.91× | 2.39±0.04 | 1.43× | 2.25±0.03 | 1.36× |
| (OnlineSPEC) **Opt-Hydra** | **2.94±0.03** | **1.90×** | **3.28±0.16** | **2.03×** | **2.71±0.05** | **1.61×** | **2.78±0.05** | **1.68×** |
| EAGLE | 1.19±0.01 | 1.00× | 1.15±0.03 | 1.00× | 1.17±0.01 | 1.00× | 1.14±0.01 | 1.00× |
| OSD-EAGLE | 1.45±0.01 | 1.18× | 1.72±0.06 | 1.46× | 1.47±0.02 | 1.28× | 1.43±0.01 | 1.20× |
| (OnlineSPEC) **Ens-EAGLE** | **1.58±0.01** | **1.33×** | **1.82±0.08** | **1.61×** | **1.62±0.01** | **1.46×** | **1.56±0.02** | **1.41×** |
| EAGLE-3 | 1.82±0.03 | 1.00× | 1.61±0.03 | 1.00× | 1.69±0.04 | 1.00× | 1.70±0.02 | 1.00× |
| OSD-EAGLE-3 | 2.25±0.02 | 1.23× | 2.42±0.12 | 1.43× | 2.09±0.10 | 1.27× | 2.13±0.02 | 1.23× |
| (OnlineSPEC) **Ens-EAGLE-3** | **2.33±0.02** | **1.27×** | **2.54±0.14** | **1.57×** | **2.18±0.10** | **1.33×** | **2.15±0.03** | **1.26×** |

**Table 2.** Evaluation of online learning-based generation-refinement methods on reasoning benchmarks. We pair each target model with a corresponding smaller draft model and report the *average accepted length* (AVGLEN ↑) with *wall-clock speedup ratio* in parentheses, and the *accuracy* (ACC (%) ↑). Results are averaged over three runs with standard deviations. The best results are highlighted in bold.

| | GSM8K | | MBPP | | MATH | | MMLU | |
|---|---|---|---|---|---|---|---|---|
| | AVGLEN ↑ | ACC (%) ↑ | AVGLEN ↑ | ACC (%) ↑ | AVGLEN ↑ | ACC (%) ↑ | AVGLEN ↑ | ACC (%) ↑ |
| | | | | Target model: *Qwen / Qwen3-8B*, Draft model: *Qwen / Qwen3-0.6B-Base* | | | | |
| Target Model | 1.00  (1.00×) | 94.32 | 1.00 (1.00×) | 53.56 | 1.00  (1.00×) | 91.54 | 1.00  (1.00×) | 83.81 |
| Draft Model | 1.00  (3.60×) | 53.84 | 1.00 (3.56×) | 14.15 | 1.00  (3.54×) | 60.66 | 1.00  (3.56×) | 55.71 |
| LR | 13.25 (1.26×) | 91.04 | 5.76 (1.09×) | 50.65 | 9.56  (1.20×) | **92.84** | 9.40  (1.21×) | 82.86 |
| OSD-LR | 12.57 (1.20×) | **93.84** | 5.66 (1.09×) | 50.54 | 6.21  (1.10×) | 89.87 | 6.67  (1.14×) | 82.14 |
| (OnlineSPEC) **Online-LR** | **14.71 (1.41×)** | 92.88 | **7.16 (1.14×)** | **51.19** | **10.63  (1.24×)** | 91.37 | **10.62  (1.26×)** | **84.52** |

correctness. Besides, we also design a naive combination of using OSD together with existing methods, including *Hydra*, *EAGLE*, *EAGLE-3*, and *LR*, and obtain *OSD-Hydra*, *OSD-EAGLE*, *OSD-EAGLE-3*, and *OSD-LR*.

**Implementation Details.** We use Vicuna-7B/13B (Chiang et al., 2023), Llama-2-7b (Touvron et al., 2023), and Qwen3-8B/32B (Yang et al., 2025) as target models. The online evaluation is conducted in a streaming fashion with chunk sizes of 40 for vanilla-SD, EAGLE, and EAGLE-3; 80 for Hydra; and 25 for Lookahead Reasoning. The offline phase uses approximately 1000 samples per domain for warm-up, followed by additional samples processed in a streaming manner during the online phase. The maximum sequence length is set to 2048 tokens, and we employ mixed-precision

training with bfloat16 to accelerate computation and reduce memory footprint. We use Flash Attention (Dao et al., 2022) to accelerate attention computation, and greedy decoding is adopted during inference across all experiments. All experiments are conducted on four NVIDIA A800 (80 GB) GPUs. More detailed implementation details are provided in the Appendix B.10.

### 4.2. Evaluation of Our Approach

In this part, to answer **Q1** and **Q2**, we evaluate OnlineSPEC on different datasets and target models.

**Update with Gradient Descent.** We evaluate the online learning approach in both regular inference tasks and rea-

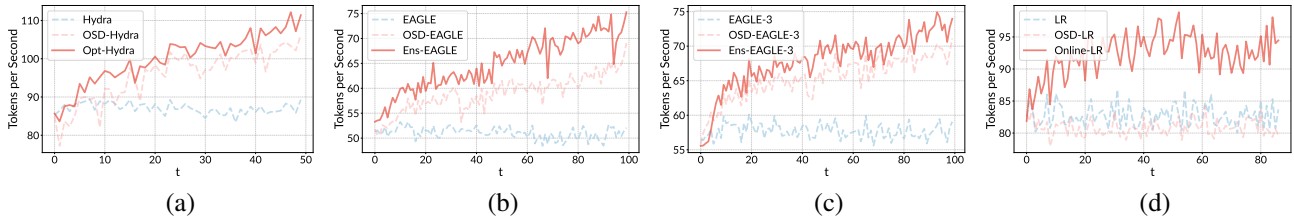

**Figure 3.** Evolution of tokens per second (TPS) on the *GSM8K* dataset: (a) *Opt-Hydra* with *lmsys/Vicuna-7B-v1.3*, (b) *Ens-EAGLE* with *lmsys/Vicuna-7B-v1.3*, (c) *Ens-EAGLE-3* with *lmsys/Vicuna-7B-v1.3*, and (d) *Online-LR* with *Qwen/Qwen3-8B*. This demonstrates consistent performance improvements via online learning, validating the effectiveness of our ONLINESPEC during deployment.

**Table 3.** Results on larger target models. We report SPEEDUP ↑ (with TPS), AVGLEN ↑, and the accuracy (ACC).

| Model | Method | SPEEDUP ↑ | AVGLEN ↑ | ACC (%) |
|---|---|---|---|---|
| Vicuna-13B | Standard AR | 1.00 (40.40) | 1.00 | – |
| | Vanilla SD | 1.20 (48.52) | 1.93 | – |
| | OSD-EAGLE-3 | 1.37 (55.52) | 2.24 | – |
| | **Ens-EAGLE-3** | **1.46** (59.03) | **2.35** | – |
| Vicuna-13B | Standard AR | 1.00 (40.40) | 1.00 | – |
| | Vanilla SD | 1.50 (60.61) | 2.22 | – |
| | OSD-Hydra | 1.69 (68.36) | 2.50 | – |
| | **Opt-Hydra** | **1.84** (74.37) | **2.73** | – |
| Qwen3-32B | Standard AR | 1.00 (35.53) | 1.00 | 96.08 |
| | LR | 1.11 (39.39) | 3.66 | 95.76 |
| | OSD-LR | 1.09 (38.63) | 3.18 | 95.44 |
| | **Online-LR** | **1.31** (46.71) | **9.98** | 95.68 |

**Table 4.** Comparison of different online learning strategies applied to Hydra on *GSM8K* with *Vicuna-13B-v1.3*.

| Method | AVGLEN ↑ | SPEEDUP ↑ |
|---|---|---|
| Hydra (baseline) | 2.22 | 1.00× (60.61) |
| OSD-Hydra | 2.50 | 1.13× (68.36) |
| Ens-Hydra (ensemble) | 2.61 | 1.15× (70.64) |
| **Opt-Hydra** (optimistic) | **2.73** | **1.23×** (74.37) |

**Table 5.** Hyperparameter analysis on *GSM8K*. We report the *average accepted length* (AVGLEN ↑), and the *wall-clock speedup ratio* (SPEEDUP ↑). *(i)* Varying learning rate on *OSD-Hydra* vs. *Opt-Hydra*. *(ii)* Varying learning rate on *OSD-EAGLE* vs *Ens-EAGLE*. *(iii)* Varying total steps $T$.

| Method | AVGLEN ↑ | SPEEDUP ↑ |
|---|---|---|
| *(i) Hydra: Varying Learning Rate vs. Opt-Hydra* | | |
| OSD-Hydra $(\eta=0.5)$ | $2.53\pm0.05$ | $1.17\pm0.02$ |
| OSD-Hydra $(\eta=0.8)$ | $2.56\pm0.06$ | $1.19\pm0.02$ |
| OSD-Hydra $(\eta=1.0)$ | $2.51\pm0.05$ | $1.16\pm0.02$ |
| OSD-Hydra $(\eta=1.5)$ | $2.27\pm0.03$ | $1.05\pm0.01$ |
| **Opt-Hydra** | $\mathbf{2.69\pm0.08}$ | $\mathbf{1.26\pm0.03}$ |
| *(ii) EAGLE: Varying Learning Rate vs. Ens-EAGLE* | | |
| OSD-EAGLE $(\eta=5\times10^{-4})$ | $1.57\pm0.03$ | $1.03\pm0.02$ |
| OSD-EAGLE $(\eta=1\times10^{-3})$ | $1.63\pm0.04$ | $1.07\pm0.03$ |
| OSD-EAGLE $(\eta=1\times10^{-2})$ | $1.92\pm0.04$ | $1.28\pm0.05$ |
| OSD-EAGLE $(\eta=1\times10^{-1})$ | $1.85\pm0.03$ | $1.22\pm0.04$ |
| **Ens-EAGLE** | $\mathbf{2.01\pm0.05}$ | $\mathbf{1.41\pm0.05}$ |
| *(iii) Opt-Hydra: Varying Total Online Steps $T$* | | |
| Opt-Hydra $(T=1000)$ | $2.33\pm0.02$ | $1.10\pm0.01$ |
| Opt-Hydra $(T=2000)$ | $2.52\pm0.05$ | $1.19\pm0.01$ |
| Opt-Hydra $(T=3000)$ | $2.55\pm0.04$ | $1.20\pm0.02$ |
| Opt-Hydra $(T=4000)$ | $\mathbf{2.69\pm0.08}$ | $\mathbf{1.26\pm0.03}$ |

soning tasks. As shown in Table 1, previous SOTA, *OSD*, achieves a better performance than the vanilla speculative decoding methods, indicating that the online update is effective for continuously improving the draft model and the inference efficiency. In our work, we further extend this idea to the reasoning tasks, as shown in Table 2, the *Online-LR* method, which conducts online update using DPO-based optimization, achieves a better performance than the offline baseline *LR* method. Note that the naive combination, *OSD-LR*, achieves a lower performance than the offline baseline *LR* method, indicating that considering token-level feedback is not suitable for reasoning tasks, further emphasizing the flexibility of our OnlineSPEC framework.

**Update with Optimistic Online Learning.** Furthermore, we evaluated the online learning approach in the Hydra framework (Ankner et al., 2024). As shown in Table 1 and Figure 3, the *Opt-Hydra* method, which conducts online update using optimistic online learning, achieves better performance than the offline baseline *Hydra* method and the contender *OSD-Hydra* method, indicating that optimistic online learning is effective for continuously improving the

draft model and reasoning efficiency due to its ability to predict the future gradient and further update the draft model.

**Update with Online Ensemble.** We also evaluate our proposed online learning approaches in the EAGLE framework (Li et al., 2024b). As demonstrated in Table 1 and Figure 3, the *Ens-EAGLE* method, inspired by the online ensemble paradigm that employs multiple draft models with different learning rates with a hedge algorithm to combine the output, achieves better performance than the offline baseline *EAGLE* method and the contender *OSD-EAGLE* method, indicating that online ensemble is effective to robustly improve the draft model during deployment.

**Additional Results on Larger LLMs.** To evaluate the scalability of our framework, we conduct additional experiments on larger target models, including *Vicuna-13B-v1.3* and *Qwen3-32B*, with results on the *GSM8K* dataset reported in Table 3. For *Vicuna-13B-v1.3*, *Ens-EAGLE-3* achieves $1.46\times$ speedup over standard AR decoding, and *Opt-Hydra* achieves $1.84\times$. For the larger *Qwen3-32B* paired with *Qwen3-1.7B* as the draft model, *Online-LR* attains $1.31\times$ speedup with an average accepted length of

9.98, substantially outperforming both *LR* and *OSD-LR* while maintaining comparable accuracy. These results confirm that OnlineSPEC scales effectively to larger models and consistently improves over baselines.

### 4.3. Hyperparameter Analysis

To answer **Q3**, as summarized in Table 5, we conduct a hyperparameter analysis of our proposed approaches.

**Effect of Learning Rate.** We first examine the impact of the learning rate $\eta$ on the performance of OSD-EAGLE and OSD-Hydra. As shown in Table 5 *(i)* and *(ii)*, using a single fixed learning rate yields inconsistent performance across different settings. Specifically, a small learning rate (e.g., $\eta = 5 \times 10^{-4}$) leads to slow adaptation, while a large learning rate (e.g., $\eta = 10^{-1}$) may cause unstable updates. Importantly, no single learning rate consistently outperforms *Opt-Hydra* or *Ens-EAGLE*, which demonstrates the effectiveness of our optimistic online learning and online ensemble approaches. This aligns with our theoretical analysis in Section 3: optimistic online learning leverages predictive hints to guide more effective updates, and ensemble learning adaptively tracks the best-performing base learner.

**Effect of Total Online Steps $T$.** We further investigate how the total number of online steps $T$ affects the performance of Opt-Hydra. As shown in Table 5 *(iii)*, the performance consistently improves as $T$ increases from 1000 to 4000. This observation is consistent with our theoretical justification in Theorem 1, where the acceleration rate $\gamma$ increases as $T$ increases. This demonstrates that OnlineSPEC can continuously leverage interactive feedback to evolve the draft model, thereby achieving better acceleration over time.

## 5. Conclusion

In this paper, we propose OnlineSPEC, a unified framework that systematically leverages interactive verification feedback to continuously evolve draft models during deployment. Different from previous advances that rely on fixed offline-trained draft models or employ task-specific design for online adaptation, we establish a principled connection between speculative decoding and online learning, and formally characterize how the algorithm's dynamic regret governs the acceleration rate, enabling systematic algorithm design by leveraging the rich toolkit from online learning. Building on OnlineSPEC, we develop three instantiations: *(i) Online-LR* with DPO-style optimization for reasoning tasks, *(ii) Opt-Hydra* with optimistic updates guided by historical gradients, and *(iii) Ens-Eagle* with an adaptive ensemble of multiple draft heads, each equipped with theoretical justifications and improved performance. Experiments on seven benchmarks and five foundation models demonstrate improvements over previous SOTA methods.

## Acknowledgements

Yu-Yang Qian, Hao-Cong Wu, and Peng Zhao were supported by NSFC (62576164) and the "111 Center" (No. B26023), and the Fundamental Research Funds for the Central Universities (2026300331). Yichao Fu and Hao Zhang were supported by UCSD HDSI. The authors would like to thank Prof. Yu-Xiang Wang for insightful and helpful discussions, and thank Yuan-Ze Xu for the early participation in the project.

## Impact Statement

This paper presents a unified framework for accelerating LLM inference through online learning, which evolves the draft models in speculative decoding. By improving the acceptance rate and reducing inference latency, our method enables more efficient utilization of computational resources, which may contribute to lower energy consumption and reduced carbon emissions in large-scale LLM serving systems, thereby promoting environmentally sustainable AI.

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

# A. Related Work

In the following, we discuss the related topics.

## A.1. Generation-Refinement Framework

This section reviews the *generation-refinement framework* (Hu et al., 2025), a broad class of LLM acceleration methods that share a common structure: a lightweight model first generates a draft sequence, which is then refined or verified by a larger target model. In the following, we introduce three typical methods within the generation-refinement paradigm: speculative decoding, cascade decoding, and multi-token prediction.

**Speculative decoding.** Speculative decoding accelerates inference in large target language models by leveraging a smaller draft model. The draft model generates a block of tokens—referred to as a draft sequence—using standard autoregressive decoding. These tokens are then verified *in parallel* by the larger model. All tokens up to the first rejected one are accepted, after which the generation reverts to that point. EAGLE (Li et al., 2024b) introduces feature-level autoregression with token-conditioned drafting to enable efficient speculative sampling. EAGLE-2 (Li et al., 2024a) builds upon EAGLE by employing a context-aware dynamic draft tree. EAGLE-3 (Li et al., 2025) further advances EAGLE-2 by replacing feature prediction with direct token prediction and leveraging multi-layer feature fusion to improve scalability. Lookahead decoding (Fu et al., 2024) accelerates LLM inference without auxiliary models by maintaining $n$-gram pools from the Jacobi decoding trajectory, which are used to generate candidate tokens for parallel verification.

On the other hand, SpecTR (Sun et al., 2023) formulates draft selection as an optimal transport problem with membership cost and proposes an algorithm for multi-candidate token-level speculation. MCSD (Yang et al., 2024) samples multiple candidate draft sequences from the draft model and organizes them into batches for parallel verification, thereby improving the acceptance rate. BanditSpec (Hou et al., 2025) casts hyperparameter selection in speculative decoding as a multi-armed bandit problem. HedgeSpec (Liu et al., 2026) avoids the bandit formulation by introducing an additional verification step that evaluates all draft models without requiring extra target model queries. For reasoning tasks, beyond token-level speculative decoding, Lookahead Reasoning (Fu et al., 2025) exploits *step-level* parallelism by having a draft model propose multiple future reasoning steps, which are then expanded in parallel by the target model and semantically verified.

However, in previous speculative decoding methods, the draft model is typically trained offline via knowledge distillation (Hinton et al., 2015). DistillSpec (Zhou et al., 2024) improves draft-target alignment through knowledge distillation by leveraging on-policy data generation and task-specific divergence functions. More recently, OSD (Liu et al., 2024b) proposes online speculative decoding, which continuously updates the draft model using observed verification feedback during deployment. ATLAS (Wang et al., 2025) further adapts to evolving workload distributions by learning from both historical patterns and live traffic, employing a dual-speculator architecture coordinated by a confidence-aware controller. DVI (Bhansali & Heck, 2025) introduces a self-speculative framework that converts verifier decisions into supervision signals for online draft model updates using reinforcement learning.

**Cascade decoding.** Cascade decoding employs a deferral policy to identify "hard" inputs, deferring them to a larger model only when necessary, while using a smaller model by default. In a typical two-model cascade, the smaller model is invoked first, and its output confidence (e.g., the probability of the generated token) is used to decide whether to defer to the larger model. Wang et al. (2022) demonstrate that cascades constructed from pre-trained models can match or exceed state-of-the-art accuracy while achieving significant speedups. Varshney & Baral (2022) introduce model cascading to jointly improve efficiency and accuracy by selectively invoking larger models based on input difficulty. FrugalGPT (Chen et al., 2024) learns an adaptive routing strategy that determines which LLMs to invoke for different queries. Narasimhan et al. (2025) combine cascade decoding with speculative execution to achieve improved cost-quality trade-offs.

**Multi-token Prediction.** Another line of research focuses on outputting multiple tokens at a time (rather than one token at a time in traditional autoregressive models), namely, multi-token prediction (MTP). These methods augment the model with additional prediction heads or modules to generate multiple candidate tokens simultaneously, which are then verified in parallel by the target model. Raj et al. (2025) apply MTP to Speech-LLaMA and reduce the number of decoder calls while maintaining recognition accuracy. Medusa (Cai et al., 2024) augments LLMs with extra decoding heads that predict multiple subsequent tokens in parallel, and employs tree-based attention to construct and verify multiple candidate continuations simultaneously. Hydra (Ankner et al., 2024) extends Medusa by introducing sequentially dependent draft heads, where each head conditions its prediction on preceding draft tokens rather than solely on verified hidden states. CLLM (Kou et al., 2024) leverages Jacobi decoding (Song et al., 2021) and trains the model to consistently predict the fixed point from any

intermediate state. Recently, DeepSeek-V3 (Liu et al., 2024a) employs sequential MTP modules that maintain a complete causal chain at each prediction depth; while primarily designed to improve training performance, these modules can be employed for speculative decoding during inference to further accelerate the inference speed. Most recently, diffusion LLMs (Nie et al., 2025; Ye et al., 2025) have emerged as a promising alternative for autoregressive (AR) models that leverage bidirectional attention to enable parallel token generation. Several recent works exploit this capability to accelerate decoding (Wu et al., 2025; Chen et al., 2025b; Qian et al., 2026).

**Comparison with existing methods.** Although previous advances that explore interactive feedback have achieved notable empirical improvements (Liu et al., 2024b; Wang et al., 2025; Bhansali & Heck, 2025), they still have several limitations. First, they rely on ad hoc algorithmic designs without principled theoretical foundations connecting their update strategies to acceleration performance. Second, they are tailored to specific scenarios: DistillSpec focuses on offline distillation with on-policy data; OSD targets token-level verification feedback; DVI explores reinforcement learning schedules for evolving draft heads; and ATLAS emphasizes system-level traffic adaptation. Consequently, they do not provide a unified framework applicable to the broader generation-refinement paradigm. In contrast, our OnlineSPEC framework addresses these gaps by formulating general generation-refinement methods as an online learning problem, for the first time, establishing a theoretical connection between algorithmic regret and acceleration rate in Theorem 1, and providing a systematic way to incorporate advanced online learning techniques within a unified formulation.

### A.2. Online Learning

In this section, we introduce the related topics of online learning.

**Online Convex Optimization.** Online learning is a powerful paradigm for timely adjusting the decision on-the-fly, which is deemed as a $T$-round iterative game between a player and an environment. At iteration $t \in \{1, \ldots, T\}$, the player selects a decision $\mathbf{w}_t$, and the environment simultaneously selects an online function $f_t$. Subsequently, the learner will suffer a loss and observe certain gradient information as the feedback. The classical measure for online learning is the *static regret* (Cesa-Bianchi & Lugosi, 2006),

$$\mathbf{Reg}_T\left(\{f_t, \mathbf{w}\}_{t=1}^T\right) \triangleq \sum_{t=1}^T f_t(\mathbf{w}_t) - \min_{\mathbf{w} \in \mathcal{W}} \sum_{t=1}^T f_t(\mathbf{w}),$$

which compares the online learner's performance against the best *fixed* decision in hindsight. The theoretical foundations of online convex optimization are well-established (Hazan, 2016). The most classical algorithm is *online gradient descent* (OGD) (Zinkevich, 2003), which updates the decision by moving along the negative gradient of the loss function. OGD achieves an $\mathcal{O}(\sqrt{T})$ static regret for convex functions, which is known to be minimax optimal. Subsequent work demonstrates that OGD is a special case of *online mirror descent* (OMD) (Orabona, 2019), which generalizes gradient descent to non-Euclidean geometries by replacing the squared Euclidean norm with a more general Bregman divergence.

With the growing interest of the community, notable progress in modern online learning has emerged along two main directions: *adaptive online learning*, which exploits environmental structure for improved performance; and *non-stationary online learning*, which handles distribution shifts in dynamic environments.

**Adaptive Online Learning.** Adaptive online learning aims to achieve better performance by exploiting predictable patterns in the environment, which reuses historical information for a better regret rate. A prominent technique is *optimistic online learning* (Rakhlin & Sridharan, 2013), which reuses historical information to construct predictive hints for future gradients. At each round, the learner performs a two-step update: one with the predicted gradient (hint) and one with the actual gradient. When the hints are accurate, this approach achieves tighter regret bounds that depend on the cumulative hint error rather than the time horizon $T$. Optimistic online learning has been successfully applied to various challenging settings, including adversarial bandits (Wei & Luo, 2018) and dynamic regret minimization (Zhao et al., 2024).

**Non-stationary Online Learning.** Non-stationary online learning has received considerable attention due to its theoretical and practical significance (Besbes et al., 2015; Zhao et al., 2025). The goal is to design algorithms capable of adapting to distribution shifts in dynamic environments. A theoretically grounded approach is *dynamic regret minimization* (Zhang et al., 2018), which evaluates performance relative to a sequence of time-varying comparators as in Eq. (2). However, optimizing the dynamic regret is challenging, since the level of environmental non-stationarity is unknown a prior, and the comparator sequence in Eq. (2) can be arbitrary.

To tackle this challenge, a principled approach is the *online ensemble* framework (Zhao et al., 2024), which employs a meta-

base two-layer structure: a pool of diverse base learners with different learning rates is maintained, and a meta-algorithm adaptively combines their outputs to track the best-performing one on the fly. Alternatively, Cutkosky (2020) and Zhang et al. (2022) employ sequential ensemble methods, in which a series of base learners are sequentially initialized and their outputs are aggregated to produce the final prediction. The online ensemble framework achieves *optimal* dynamic regret without prior knowledge of environmental non-stationarity. Several algorithms have emerged from this framework to handle various types of distribution shift.

# B. Additional Experimental Results

In this section, we provide additional experimental results.

## B.1. Detailed Results

In this section, we provide detailed experimental results by presenting temporal evolution plots for all methods across different benchmark datasets. Each figure group compares the baseline method with its OSD-enhanced variant and our proposed OnlineSPEC approach, demonstrating the performance improvements in terms of both average accepted length and wall-clock speedup ratio as inference progresses.

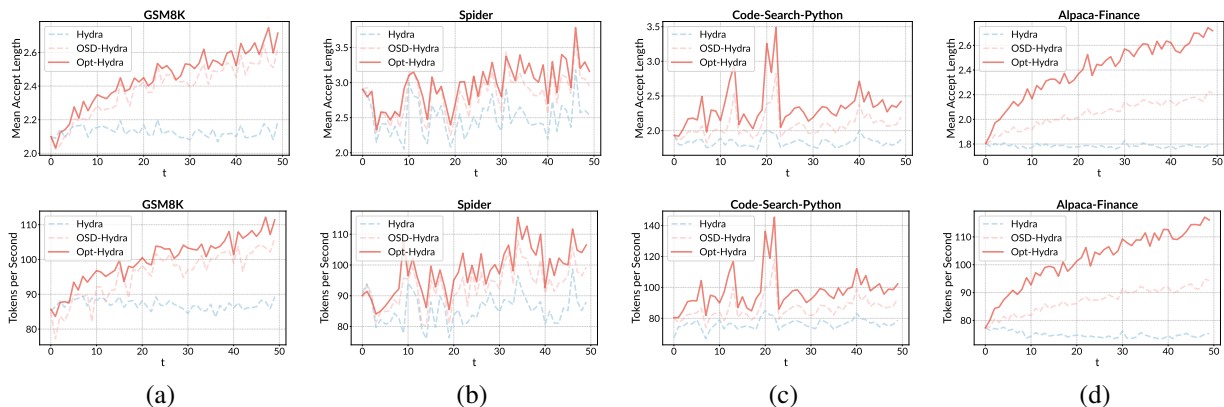

**Figure 4.** Performance comparison of *Hydra*, *OSD-Hydra*, and *Opt-Hydra* on (a) *GSM8K*, (b) *Spider*, (c) *Code-Search*, and (d) *Alpaca-Finance* using *lmsys/Vicuna-7B-v1.3* as the foundation model. We report the *average accepted length* (AVGLEN, top row) and *tokens per second* (TPS, bottom row) as inference evolves over time.

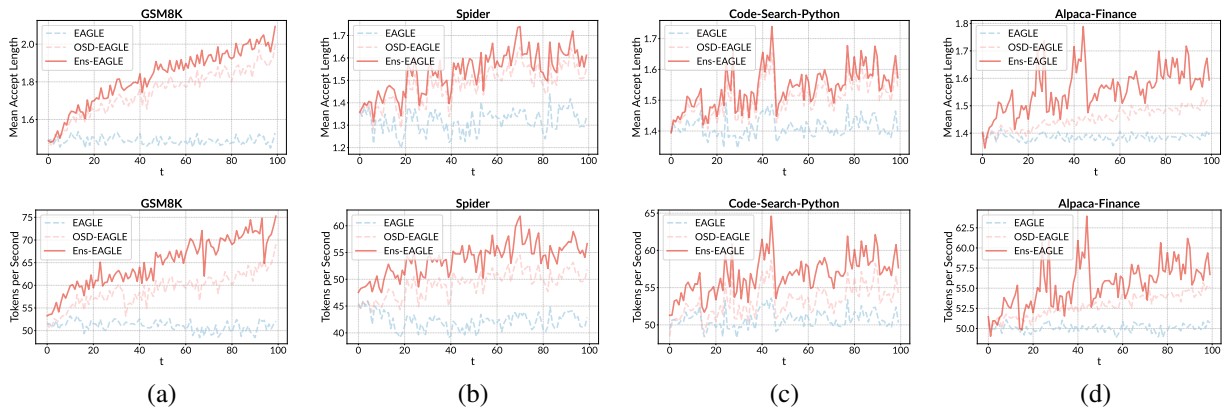

**Figure 5.** Performance comparison of *EAGLE*, *OSD-EAGLE*, and *Ens-EAGLE* on (a) *GSM8K*, (b) *Spider*, (c) *Code-Search*, and (d) *Alpaca-Finance* using *lmsys/Vicuna-7B-v1.3* as the foundation model. We report the *average accepted length* (AVGLEN, top row) and *tokens per second* (TPS, bottom row) as inference evolves over time.

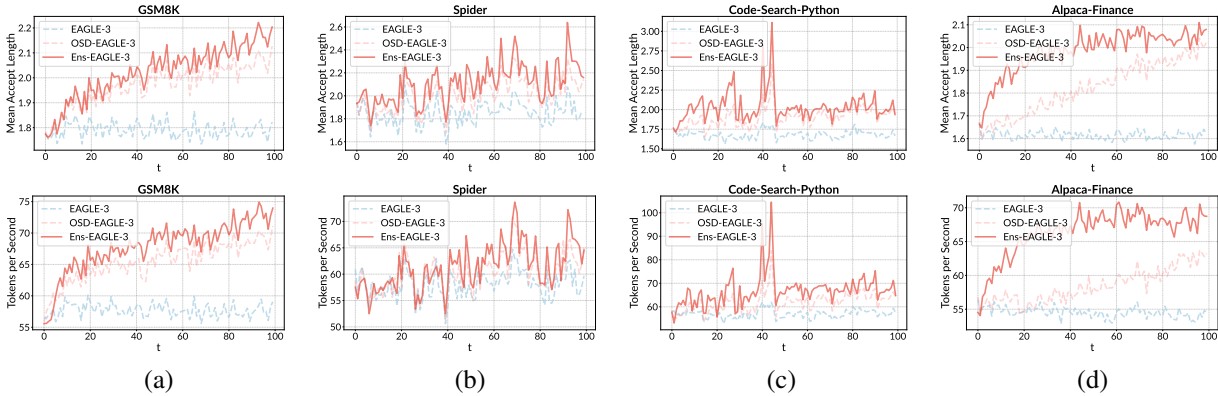

**Figure 6.** Performance comparison of *EAGLE-3*, *OSD-EAGLE-3*, and *Ens-EAGLE-3* on (a) *GSM8K*, (b) *Spider*, (c) *Code-Search*, and (d) *Alpaca-Finance* using *lmsys/Vicuna-7B-v1.3* as the foundation model. We report the *average accepted length* (AVGLEN, top row) and *tokens per second* (TPS, bottom row) as inference evolves over time.

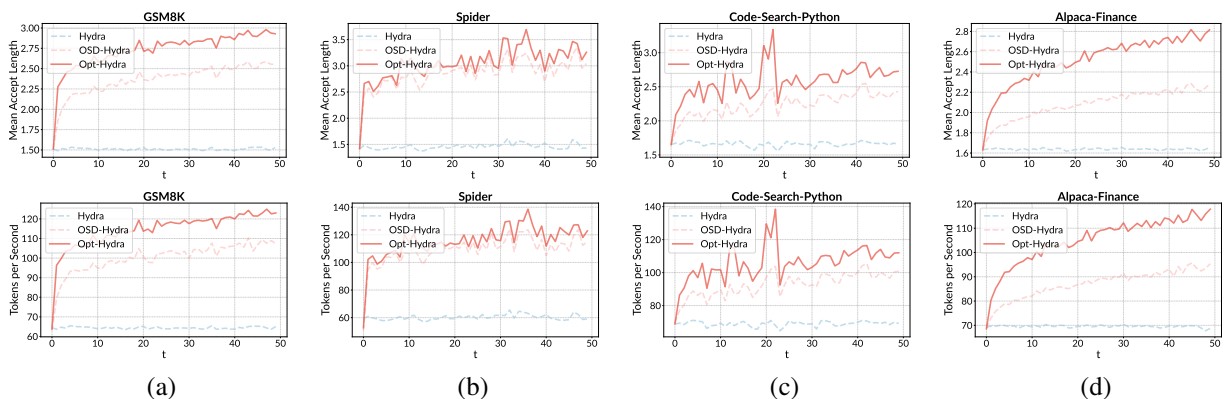

**Figure 7.** Performance comparison of *Hydra*, *OSD-Hydra*, and *Opt-Hydra* on (a) *GSM8K*, (b) *Spider*, (c) *Code-Search*, and (d) *Alpaca-Finance* using *meta-llama/Llama-2-7B-Chat* as the foundation model. We report the *average accepted length* (AVGLEN, top row) and *tokens per second* (TPS, bottom row) as inference evolves over time.

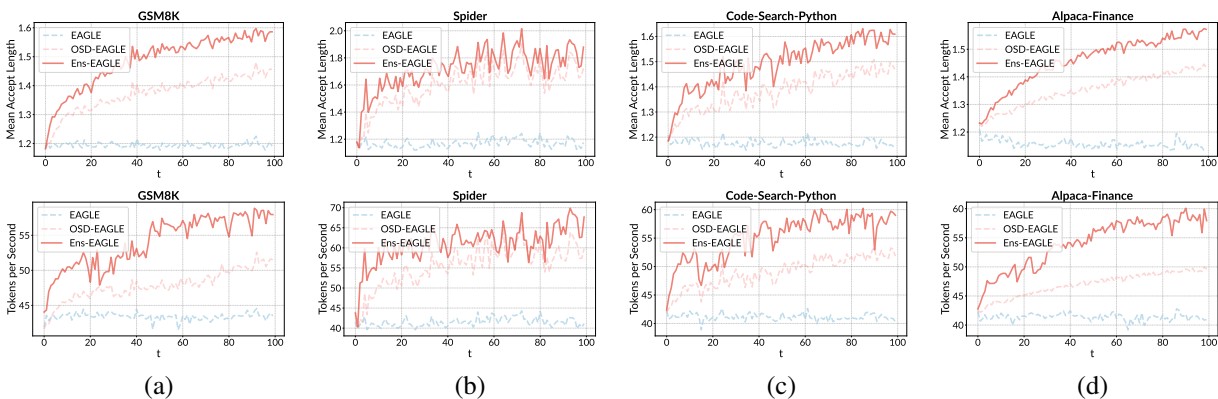

**Figure 8.** Performance comparison of *EAGLE*, *OSD-EAGLE*, and *Ens-EAGLE* on (a) *GSM8K*, (b) *Spider*, (c) *Code-Search*, and (d) *Alpaca-Finance* using *meta-llama/Llama-2-7B-Chat* as the foundation model. We report the *average accepted length* (AVGLEN, top row) and *tokens per second* (TPS, bottom row) as inference evolves over time.

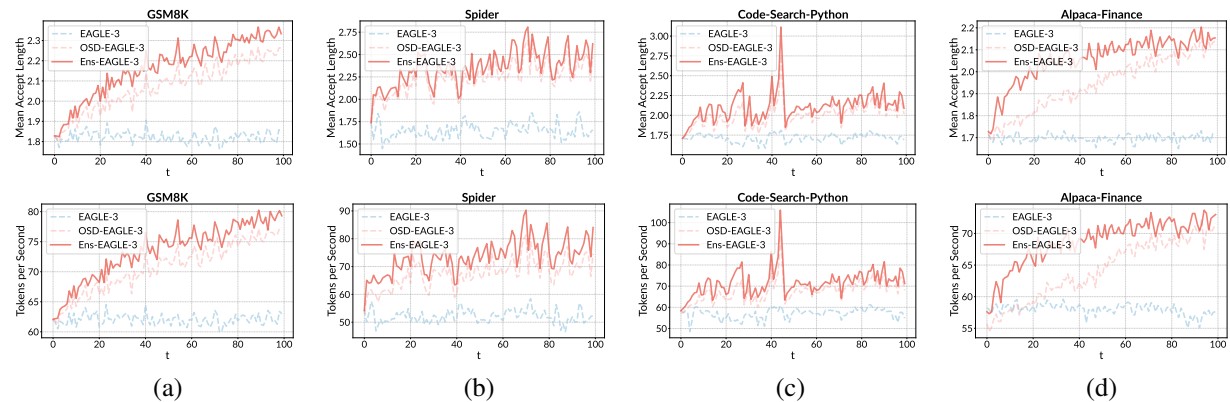

**Figure 9.** Performance comparison of *EAGLE-3*, *OSD-EAGLE-3*, and *Ens-EAGLE-3* on (a) *GSM8K*, (b) *Spider*, (c) *Code-Search*, and (d) *Alpaca-Finance* using *meta-llama/Llama-2-7B-Chat* as the foundation model. We report the *average accepted length* (AVGLEN, top row) and *tokens per second* (TPS, bottom row) as inference evolves over time.

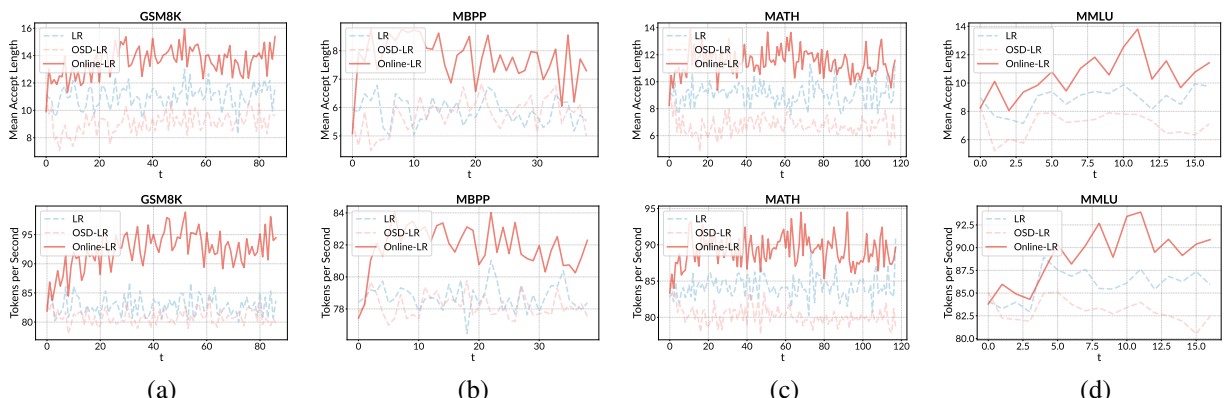

**Figure 10.** Performance comparison of *LR*, *OSD-LR*, and *Online-LR* on (a) *GSM8K*, (b) *MBPP*, (c) *MATH*, and (d) *MMLU* using *Qwen/Qwen3-8B* paired with *Qwen3-0.6B-Base* as the draft model. We report the *average accepted length* (AVGLEN, top row) and *tokens per second* (TPS, bottom row) as inference evolves over time.

### B.2. Considering Training Overhead

A natural concern regarding online updates in the OnlineSPEC framework is the additional computational cost introduced by training updates. In practice, however, this overhead can be effectively mitigated. Specifically, the online update of the draft model can be performed asynchronously on separate devices (e.g., using separate training GPUs), decoupled from the inference pipeline. Under such a deployment, the training process runs in parallel without blocking or slowing down the inference service, and the updated draft model parameters can be periodically synchronized to the inference server.

**Table 6.** Comparison of inference time vs. our online training + inference time. Our methods (Opt-Hydra and Ens-EAGLE-3) achieve better speedup even when accounting for training overhead.

| Method | Code-Search | | Alpaca-Finance | |
|---|---|---|---|---|
| | TIME (s) ↓ | SPEEDUP ↑ | TIME (s) ↓ | SPEEDUP ↑ |
| EAGLE-3 | 300.83 | 1.00 | 347.57 | 1.00 |
| **Ens-EAGLE-3** (+train) | **280.23** | **1.07** | **306.31** | **1.13** |
| Hydra | 527.46 | 1.00 | 393.18 | 1.00 |
| **Opt-Hydra** (+train) | **515.00** | **1.02** | **371.37** | **1.05** |

To further quantify the practical impact, we conduct an additional experiment where we measure the total wall-clock time, including both training and inference. As shown in Table 6, even when accounting for the training overhead, our methods still achieve notable speedups over the baselines. For instance, *Ens-EAGLE-3* achieves speedup on *Code-Search* and *Alpaca-Finance* datasets, respectively, demonstrating that the efficiency gains from improved draft model quality outweigh the additional training cost. These results suggest that the training overhead is not a practical bottleneck for deploying our online learning framework.

### B.3. Comparison with Standard AR Decoding

To further demonstrate the efficiency gains, we provide a comparison against standard autoregressive (AR) decoding. We measure the wall-clock throughput in tokens per second (TPS) on the *GSM8K* dataset across three target models. As shown in Table 7, both *Ens-EAGLE-3* and *Opt-Hydra* achieve substantial throughput improvements over standard AR decoding across all three target models.

**Table 7.** Comparison against standard AR decoding on *GSM8K*. We report TPS ↑ and relative speedup.

| Method | Vicuna-7B | Llama-2-7B | Vicuna-13B |
|---|---|---|---|
| Standard AR | 54.11 | 53.76 | 40.40 |
| **Ens-EAGLE-3** | 73.96 (+36.7%) | 79.30 (+47.5%) | 59.03 (+46.1%) |
| **Opt-Hydra** | **107.17** (+98.1%) | **122.61** (+128.1%) | **74.37** (+84.1%) |

Notably, *Opt-Hydra* attains up to $2.28\times$ the throughput of standard AR decoding on *Llama-2-7B-Chat* (+128.1%), while *Ens-EAGLE-3* consistently delivers over 36% speedup. These gains are also observed on the larger *Vicuna-13B-v1.3* model, where *Opt-Hydra* achieves 84.1% improvement, demonstrating that the effectiveness of our OnlineSPEC framework.

### B.4. Illustration of Theoretical Trends

We further provide an empirical illustration of the theoretical bounds, where we take *Ens-EAGLE* as an example. We approximate the path length $P_T$ by measuring the cumulative $\ell_2$-distance between consecutive prompts' embeddings as a proxy for environmental non-stationarity. As illustrated in Figure 11, the observed speedup increases steadily as online learning progresses, exhibiting a trend qualitatively consistent with the theoretical prediction in Corollary 3 that the acceleration rate improves as $(1 + P_T)^{1/4}/T^{1/4}$ decreases. This confirms that our theoretical analysis provides meaningful guidance for the behavior of the OnlineSPEC framework in practice.

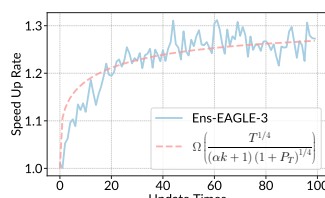

**Figure 11.** Theoretical trend vs. observed speedup for *Ens-EAGLE* on *GSM8K* with *Vicuna-7B-v1.3*.

### B.5. Sensitivity Analysis of Warm-Up Samples

We analyze the sensitivity of the number of offline warm-up samples. As shown in Table 8, *Ens-EAGLE* consistently outperforms *OSD-EAGLE* across all warm-up sizes. Notably, the performance gap widens with more warm-up samples, as stronger initialization yields more diverse base learners that the ensemble can more effectively combine.

**Table 8.** Sensitivity to the number of offline warm-up samples on *GSM8K* dataset with *Vicuna-7B-v1.3*.

| Warm-up | AVGLEN ↑ | | TPS ↑ | |
|---|---|---|---|---|
| | OSD-EAGLE | **Ens-EAGLE** | OSD-EAGLE | **Ens-EAGLE** |
| 500 | 1.75 | **1.78** | 63.28 | **65.52** |
| 1000 | 1.88 | **1.91** | 69.66 | **70.82** |
| 1500 | 2.03 | **2.17** | 76.13 | **80.07** |

### B.6. Combination of Particular Speculative Methods

The pairing of specific online learning strategies with particular speculative decoding methods is motivated by structural compatibility. Hydra (Ankner et al., 2024) employs sequentially dependent draft heads that exploit contextual dependencies across positions, which naturally aligns with optimistic online learning that reuses historical gradients as predictive hints. EAGLE (Li et al., 2024b) constructs tree-structured drafts with multiple candidate branches, which naturally corresponds to ensemble learning that maintains and combines diverse base learners. To validate these design choices, we compare alternative pairings on

**Table 9.** Comparison of different online learning strategies applied to Hydra on *GSM8K* with *Vicuna-13B-v1.3*.

| Method | AVGLEN ↑ | SPEEDUP ↑ |
|---|---|---|
| Hydra (baseline) | 2.22 | $1.00\times$ (60.61) |
| OSD-Hydra | 2.50 | $1.13\times$ (68.36) |
| Ens-Hydra (ensemble) | 2.61 | $1.15\times$ (70.64) |
| **Opt-Hydra** (optimistic) | **2.73** | **$1.23\times$** (74.37) |

*GSM8K* with *Vicuna-13B-v1.3*. As shown in Table 9, while both strategies improve over the offline baseline, the structurally aligned pairing (*Opt-Hydra*) consistently achieves the best performance.

### B.7. Details of Contenders

In this part, we provide more details about the contenders used in our experiments.

- *Vanilla speculative decoding* (Leviathan et al., 2023; Chen et al., 2023) uses a small draft model to propose candidate token continuations, which are then verified in parallel by the target model. This approach accelerates inference by reducing the number of sequential decoding steps while preserving the output distribution of the target model.

- *OSD* (Liu et al., 2024b) extends vanilla speculative decoding by periodically updating the draft model using observed feedback via knowledge distillation. This online adaptation allows the draft model to better align with the target model's

distribution over time.

- *Hydra* (Ankner et al., 2024) improves draft head-based speculation by introducing sequential dependency among draft tokens. Unlike standard draft heads that predict tokens independently, Hydra heads condition each speculation on preceding tokens in the candidate continuation, significantly improving draft accuracy. The draft model is a HydraPrefixMLP consisting of a prefix embedding layer (one complete LlamaDecoder layer), four grounded MLP heads with progressively increasing input dimensions, and four language modeling heads, totaling approximately 1.53B parameters.

- *EAGLE* (Li et al., 2024b) performs autoregression at the feature level, predicting second-to-top-layer features with a lightweight draft head to enable efficient token speculation. The draft model comprises a single Transformer layer with input and output projection layers ($\sim$0.24B parameters).

- *EAGLE-3* (Li et al., 2025) further improves EAGLE by replacing feature prediction with direct token prediction and employing multi-layer feature fusion, enabling better scalability and performance. The draft model follows a similar architecture to EAGLE but with increased capacity ($\sim$0.5B parameters).

- *LR* (Lookahead Reasoning) (Fu et al., 2025) exploits step-level parallelism for reasoning models by proposing multiple future reasoning steps simultaneously. A lightweight draft model generates step proposals, the target model expands each proposal in one batched pass, and a verifier retains semantically correct steps. This approach combines with token-level speculative decoding to achieve multiplicative speedups. We use *Qwen3-0.6B-Base* (0.6B parameters) as the draft model.

- *OSD-Hydra*, *OSD-EAGLE*, *OSD-EAGLE-3*, and *OSD-LR* are naive combinations that apply the online distillation mechanism from OSD to the Hydra, EAGLE, and LR frameworks, respectively. These baselines serve to evaluate whether simple online adaptation can improve the performance of existing speculative decoding methods.

## B.8. Details of Datasets

We evaluate our proposed approach on the following widely-used benchmark datasets:

- *GSM8K* (Cobbe et al., 2021) is a dataset of 8.5K high-quality, linguistically diverse grade school math word problems. Each problem requires 2 to 8 steps of multi-step reasoning using basic arithmetic operations ($+$, $-$, $\times$, $\div$) to derive the final answer, with solutions provided in natural language.

- *Spider* (Yu et al., 2019) is a large-scale, cross-domain semantic parsing and text-to-SQL dataset containing 10,181 questions and 5,693 unique complex SQL queries across 200 databases spanning 138 domains.

- *Code-search-Python* (Husain et al., 2020) is a subset of the CodeSearchNet corpus that focuses on Python code retrieval using natural language queries. It provides paired data of natural language descriptions and corresponding Python code snippets, enabling evaluation of semantic code search capabilities.

- *Alpaca-finance* (Taori et al., 2023) is a domain-specific instruction-following dataset derived from the Stanford Alpaca framework, tailored for financial applications. It contains instruction-response pairs designed to evaluate model performance on finance-related tasks.

- *MBPP* (Austin et al., 2021) (Mostly Basic Programming Problems) is a dataset of 974 Python programming problems designed to evaluate the problem-solving capabilities of language models. Each problem includes a natural language description, a reference solution, and test cases for validation.

- *MATH* (Lewkowycz et al., 2022) is a dataset of 12,500 challenging competition-level mathematics problems covering a wide range of topics such as algebra, geometry, calculus, and number theory. Each problem is accompanied by a detailed solution written in natural language and LaTeX.

- *MMLU* (Hendrycks et al., 2021) (Massive Multitask Language Understanding) is a benchmark designed to evaluate a model's multitask accuracy across 57 tasks spanning STEM, humanities, social sciences, and other professional fields. It tests both in-domain and out-of-domain generalization capabilities.

## B.9. Prompt Templates

We provide the prompt templates used in our experiments. For the reasoning framework (Online-LR), we employ Qwen2.5-7B-Instruct (Qwen et al., 2025) as the judge model to verify semantic alignment between the draft model's predictions and the target model's outputs. The judge model evaluates whether two reasoning steps convey the same meaning, focusing on semantic similarity rather than exact wording. Table 10 presents the prompt template used for the alignment verification.

**Table 10.** Prompt template used for semantic alignment verification in reasoning. We use *Qwen2.5-7B-Instruct* as judge (temperature=0.0).

---

**System Prompt:**
You are Qwen, created by Alibaba Cloud. You are a helpful assistant.

**User Prompt:**
Evaluate whether the following two reasoning steps (s1 and s2) convey exactly the same meaning. Focus on semantic similarity rather than exact wording.
Compare the main ideas, key points, overall message, logical structure, and numerical calculations/results of both reasoning steps.
If the reasoning steps convey essentially the same meaning and generate same calculation results, respond with [aligned]. If the reasoning steps express different meanings, respond with [unaligned]. If it is too hard to determine, respond with [unaligned]
Please directly provide the final result in [aligned] or [unaligned].
Reasoning step 1 (s1):
<start_s1>
{draft_output}
<end_s1>
Reasoning step 2 (s2):
<start_s2>
{target_output}
<end_s2>

---

**Table 11.** Problem-solving prompt template for math reasoning tasks.

---

**User Prompt:**
{question}
Please reason step by step, and put your final answer within \boxed{}. Once a full round of reasoning is completed, do not check again, immediately cease reasoning, and output the answer.

---

For the problem-solving prompt in reasoning, we use the prompt template shown in Table 11 to instruct the reasoning model.

To prevent excessive reasoning behavior, we inject a transition instruction into the context when the token count reaches the predefined limit of 1024 tokens, as shown in Table 12:

**Table 12.** Thinking-to-Output mode transition instruction.

---

**Context:** {preceding prompt and reasoning trace}
**Instruction:** Considering the limited time by the user, I have to give the solution based on the thinking directly now.
</think>

---

### B.10. More Implementation Details

Throughout experiments, we utilize three foundation models as the target model in the generation-refinement framework, including Vicuna-7B (Chiang et al., 2023), Llama-2-7b (Touvron et al., 2023), and Qwen3-8B (Yang et al., 2025). The online evaluation is conducted in a streaming fashion: we partition the evaluation dataset into chunks of size $T = 40$ for the EAGLE framework, $T = 80$ for Hydra, and $T = 25$ for reasoning tasks, with total iterations ranging from 16 to 120 depending on the dataset size. The maximum sequence length is set to 2048 tokens. We use Flash Attention to accelerate attention computation. The offline training phase uses approximately 1000 samples per domain for warm-up, and the online phase processes approximately 4000 additional samples in a streaming manner. We employ mixed-precision training with bfloat16 (bf16) to accelerate computation and reduce memory footprint. All experiments were performed using four NVIDIA A800 (80 GB) GPUs with two Intel(R) Xeon(R) Gold 6430 CPUs.

For the online gradient descent experiments (Online-LR), we use the AdamW optimizer with $\beta_1 = 0.9$, $\beta_2 = 0.95$, and

gradient clipping set to $1.0$. DPO-based updates are performed with a learning rate of $5 \times 10^{-7}$, a preference scaling parameter $\beta = 0.1$, and are trained for 3 epochs per online update round. SFT-based updates use the same learning rate of $5 \times 10^{-7}$ with a constant warmup schedule for one epoch. The global batch size is set to 16, with a micro-batch size of 2 per GPU. During inference-time sampling, the temperature is fixed at $t = 0.6$, and a repetition penalty of $1.2$ is applied to mitigate repetitive outputs, which could otherwise degrade the quality of the collected training data. To prevent excessive reasoning behavior, we restrict the thinking budget to 1024 tokens; once this limit is reached, the model terminates the thinking process and switches to output generation.

For the EAGLE experiments, we employ the Adam optimizer with $\beta_1 = 0.9$ and $\beta_2 = 0.95$, using a learning rate of $3 \times 10^{-5}$. Training is conducted for 5 epochs per online iteration, with gradient clipping applied at a threshold of $0.5$. In the online ensemble setting, we maintain $N = 3$ base learners (draft models) with geometrically spaced learning rates $\{3 \times 10^{-5}, 6 \times 10^{-5}, 1.2 \times 10^{-4}\}$. All draft models can be updated in parallel within each iteration, incurring no additional training overhead compared to a single draft model update. The meta-learner is formed via an exponentially weighted sum of the base learners. The new weights of meta learner is updated as $\mathbf{w}_t = \sum_{i=1}^{3} p_t^i \cdot \mathbf{w}_t^i$, where $p_t^i \propto \exp(-\varepsilon \sum_{s=1}^{t-1} f_t(\mathbf{w}_s^i))$, $f_t(\cdot)$ is the training loss of iteration $t$. The parameter $\varepsilon$ is set as 10. The EAGLE-3 configuration follows the same setup as EAGLE, except that the number of training epochs per iteration is reduced to 2, and the learning rates for the online ensemble are adjusted to $\{1 \times 10^{-4}, 2 \times 10^{-4}, 4 \times 10^{-4}\}$. For the Hydra framework, we use SGD with a momentum of $0.9$, a learning rate of $0.1$, and a dropout rate of $0.1$. Across all experiments, greedy decoding is adopted during inference.

# C. Other Applications

In this section, we further showcase the other applications of our framework.

## C.1. Dynamically Determine the Candidate Length

We further remark that our framework can be used to determine the candidate length $k$ of the draft model.

**Corollary 4.** For a draft model with average accepted rate $\text{Acc}_t$ and inference time ratio $\alpha$ relative to the target model, the optimal candidate length for the time step $t$ is

$$k = \Theta\left(\frac{1}{\alpha(1 - \text{Acc}_t)}\right).$$

*Proof of Corollary 4.* As demonstrated in Theorem 1,

$$\mathbb{E}[n_t] = \frac{1 - \text{Acc}_t^k}{1 - \text{Acc}_t}$$

Therefore,

$$\gamma_t = \frac{1 - \text{Acc}_t^k}{(1 - \text{Acc}_t)(\alpha k + 1)}$$

Since the original equation has the term $\text{Acc}_t^k$, we can not solve it directly as it is a transcendental equation. Instead, by Taylor expansion, we have

$$\text{Acc}_t^k \approx \text{Acc}_t^k \leq 1 + k(\text{Acc}_t - 1) + \frac{k(k-1)}{2}(\text{Acc}_t - 1)^2.$$

where the error term is bounded by $O\left(|1 - \text{Acc}_t|^3\right)$. Therefore, by solving the optimal value of $k$

$$\partial_k \gamma_t \approx \frac{k - \text{Acc}_t k - \frac{1}{2}(1 - \text{Acc}_t)^2(-1 + k)k}{(1 - \text{Acc}_t)(1 + \alpha k)}$$

By solving the equation $\partial_k \gamma_t = 0$, we have

$$k = \frac{\mathcal{C}}{\alpha(1 - \text{Acc}_t)}$$

where $\mathcal{C} \triangleq -1 + \text{Acc}_t + \sqrt{1 - 2\text{Acc}_t + \text{Acc}_t^2 + 3\alpha - 4\text{Acc}_t\alpha + \text{Acc}_t^2\alpha} \in [0, 1]$ is a scale factor. Therefore, the optimal candidate length is $k = \Theta\left(\frac{1}{\alpha(1 - \text{Acc}_t)}\right)$. $\square$

This corollary provides a principled approach to determining the optimal candidate length following theoretical guidance. It demonstrates that the candidate length should be adjusted based on both the draft model's accuracy (reflected in $\mathrm{Acc}_t$) and the computational efficiency ratio between the draft and target models (reflected in $\alpha$). When the draft model is highly accurate (small $\mathrm{Acc}_t$) or computationally efficient relative to the target model (small $\alpha$), a larger candidate length can be used to maximize acceleration rate.

### C.2. Bandit Online Learning

In the main text, we focus on the full-information online learning setting, where the player can observe the complete loss function, including its gradients. However, in certain scenarios, the learner may only have access to partial feedback, such as the player can only observe loss value at a certain queried draft model. This motivates the study of *bandit online learning*.

**Bandit Techniques in Speculative Decoding.** Recent work has explored bandit techniques to improve speculative decoding. BanditSpec (Hou et al., 2025) formulates the hyperparameter selection problem (e.g., candidate length, draft model choice) as a multi-armed bandit problem. Specifically, it treats each hyperparameter configuration as an arm and uses the acceptance rate as the reward signal. Two bandit-based algorithms, UCBSpec and EXP3Spec, are proposed to adaptively select hyperparameters during text generation, achieving near-optimal stopping time regret under both stochastic and adversarial reward settings. MetaSD (Kim et al., 2025) tackles the limitation of single-drafter approaches by incorporating multiple draft models into the speculative decoding process. It employs multi-armed bandit sampling to allocate computational resources across different drafters based on their observed acceptance rates, thereby improving overall generation performance.

**Incorporating Bandit Techniques into the OnlineSPEC Framework.** We remark that our OnlineSPEC framework can naturally incorporate bandit online learning techniques to *continuously* improve the draft sequence quality during deployment. In the deployment scenario, the verification outcome (accept/reject) from the target model serves as a natural reward signal that can be exploited by bandit algorithms, i.e., modify the update step in Algorithm 1 to use bandit feedback. Specifically, one can maintain a pool of candidate draft models (or hyperparameter configurations) and use bandit algorithms such as UCB or EXP3 to adaptively select the best-performing option at each step based on the observed acceptance rates. As more queries are processed, the bandit algorithm accumulates feedback and progressively identifies the optimal draft model, leading to continuous performance improvement over time. We leave this extension as future work.

### C.3. Combining Optimistic Online Learning and Ensemble Learning

In Section 3.2 and Section 3.3, we have demonstrated that our OnlineSPEC framework can be instantiated with optimistic online learning and ensemble learning, respectively. Specifically, optimistic learning exploits predictive hints (e.g., historical gradients) to achieve improved regret when hints are accurate (Corollary 2), while ensemble learning hedges against non-stationarity by maintaining multiple base learners with different adaptation rates (Corollary 3). For simplicity and clarity, we consider each technique in isolation in the main paper.

It is worth noting that these two approaches can be combined together to leverage the benefits of both. Recent advances in online learning, Sword++ (Zhao et al., 2024), demonstrate that integrating optimistic updates into the ensemble framework where both the meta-algorithm and base learners employ optimistic mirror descent. Such a combination would enable the draft model to both exploit temporal locality via optimism and hedge against abrupt distribution shifts via ensemble, potentially leading to further performance improvements. Nevertheless, incorporating both techniques into the OnlineSPEC framework may bring additional algorithm complexity, we leave this extension as future work.

### C.4. Speculative Decoding in On-Policy Reinforcement Learning

Recent works begin to explore the application of speculative decoding in reinforcement learning (Chen et al., 2025a; Liu et al., 2025; Chang et al., 2026). Specifically, in RL training, the rollout phase, in which the policy generates trajectories for learning, constitutes a major computational bottleneck. Speculative decoding offers a promising approach to accelerate this phase by accelerating the generate speed. However, a key challenge arises in on-policy settings: the policy model continuously evolves during training, causing a static draft model to become increasingly misaligned with the target policy over time. This distribution drift leads to lower acceptance rates and diminished speedup as training progresses. Our OnlineSPEC framework is naturally suited to address this challenge, as it enables the draft model to continuously adapt to the evolving target policy through interactive feedback.

# D. Proofs

In this section, we provide the omitted proofs for the theoretical justifications presented in Section 2 and Section 3.

## D.1. Proof of Lemma 1

*Proof.* We first consider the expected length of accepted tokens at step $t$. Following Leviathan et al. (2023), we make the simplifying assumption of Assumption 1 that the conditional distributions $q_{\mathbf{w}_t}(x \mid \mathbf{x}_{<i})$ follow an *i.i.d.* distribution for any $i \in \{1, \ldots, k\}$, and similarly for $p_{\mathbf{v}}(x \mid \mathbf{x}_{<i})$. Recall that in Algorithm 1, the draft model $\mathbf{w}_t$ generates the draft sequence from the distribution $\{q_{\mathbf{w}_t}(x \mid \mathbf{x}_{<i})\}_{i=1}^k$, and the target model $\mathbf{v}$ verifies them using the distribution $\{p_{\mathbf{v}}(x \mid \mathbf{x}_{<i})\}_{i=1}^k$, where $\mathbf{x}$ denotes the context sequence. At time step $t$, according to the rejection sampling criterion as in Algorithm 1 (accept the token only when $r_j \leq \frac{p_{\mathbf{v}}(x_j \mid \mathbf{x}_{<j})}{q_{\mathbf{w}_t}(x_j \mid \mathbf{x}_{<j})}$), the expected acceptance rate of the draft model is given by:

$$\mathrm{Acc}_t \triangleq \mathbb{E}_{x \sim q_{\mathbf{w}_t}(\cdot \mid \mathbf{x})} \left[ \min \left\{ 1, \frac{p_{\mathbf{v}}(x \mid \mathbf{x})}{q_{\mathbf{w}_t}(x \mid \mathbf{x})} \right\} \right] = \sum_x \min \left\{ p_{\mathbf{v}}(x \mid \mathbf{x}), q_{\mathbf{w}_t}(x \mid \mathbf{x}) \right\}$$

$$= 1 - \frac{1}{2} \sum_x |p_{\mathbf{v}}(x \mid \mathbf{x}) - q_{\mathbf{w}_t}(x \mid \mathbf{x})| = 1 - \mathrm{TV}\left( p_{\mathbf{v}}(\cdot \mid \mathbf{x}), q_{\mathbf{w}_t}(\cdot \mid \mathbf{x}) \right),$$

where the third equality follows from the identity $\min(a, b) = \frac{1}{2}(a + b - |a - b|)$ and the assumption that both $p_{\mathbf{v}}(\cdot \mid \mathbf{x})$ and $q_{\mathbf{w}_t}(\cdot \mid \mathbf{x})$ are probability distributions summing to one. Here, $\mathrm{TV}(\cdot, \cdot) : \mathcal{D} \times \mathcal{D} \mapsto \mathbb{R}$ denotes the total variation distance. Additionally, since we choose the cross-entropy loss, i.e.,

$$f_t(\mathbf{w}_t) \triangleq -\mathbb{E}_{x \sim p_{\mathbf{v}}(\cdot \mid \mathbf{x})} \left[ \log q_{\mathbf{w}_t}(x \mid \mathbf{x}) \right],$$

and compare with optimal draft model $\mathbf{w}_t^\star$ that perfectly matches the target distribution, i.e., $q_{\mathbf{w}_t^\star}(\cdot \mid \mathbf{x}) = p_{\mathbf{v}}(\cdot \mid \mathbf{x})$, we have

$$f_t(\mathbf{w}_t) - f_t(\mathbf{w}_t^\star) = \mathbb{E}_{x \sim p_{\mathbf{v}}(\cdot \mid \mathbf{x})} \log \frac{p_{\mathbf{v}}(x \mid \mathbf{x})}{q_{\mathbf{w}_t}(x \mid \mathbf{x})} = \mathrm{KL}\left( p_{\mathbf{v}}(\cdot \mid \mathbf{x}) \, \| \, q_{\mathbf{w}_t}(\cdot \mid \mathbf{x}) \right). \tag{4}$$

Following Leviathan et al. (2023), the expected number of output tokens generated at step $t$ is:

$$\mathbb{E}[n_t] = \frac{1 - \mathrm{Acc}_t^{k+1}}{1 - \mathrm{Acc}_t}.$$

To derive a lower bound on $\mathbb{E}[n_t]$, we first upper bound $\mathrm{Acc}_t^{k+1}$. Since $\ln a \leq -(1 - a)$ for all $a \in (0, 1)$, we have

$$\mathrm{Acc}_t^{k+1} = e^{(k+1) \ln \mathrm{Acc}_t} \leq e^{-(k+1)(1 - \mathrm{Acc}_t)} = e^{-(k+1) \mathrm{TV}_t},$$

where $\mathrm{TV}_t \triangleq \mathrm{TV}(p_{\mathbf{v}}(\cdot \mid \mathbf{x}), q_{\mathbf{w}_t}(\cdot \mid \mathbf{x}))$. Substituting into the expression for $\mathbb{E}[n_t]$ yields

$$\mathbb{E}[n_t] \geq \frac{1 - e^{-(k+1) \mathrm{TV}_t}}{\mathrm{TV}_t}.$$

We further simplify this bound using the elementary inequality $1 - e^{-u} \geq \frac{u}{1+u}$ for all $u > 0$, which follows directly from the convexity bound $e^u \geq 1 + u$. Applying this with $u = (k+1) \mathrm{TV}_t$ gives

$$\mathbb{E}[n_t] \geq \frac{k+1}{1 + (k+1) \mathrm{TV}_t}.$$

Note that this lower bound is always at most $k + 1$, consistent with the natural upper bound $\mathbb{E}[n_t] \leq k + 1$.

Summing over $t = 1, \ldots, T$ and applying the Cauchy-Schwarz inequality ($\sum_{t=1}^T \frac{1}{a_t} \geq \frac{T^2}{\sum_{t=1}^T a_t}$) with $a_t = 1 + (k+1) \mathrm{TV}_t$, we obtain

$$\mathbb{E}[|\hat{\mathbf{x}}|] = \sum_{t=1}^T \mathbb{E}[n_t] \geq (k+1) \cdot \frac{T^2}{\sum_{t=1}^T \left( 1 + (k+1) \mathrm{TV}_t \right)} = \frac{(k+1) T^2}{T + (k+1) \sum_{t=1}^T \mathrm{TV}_t}. \tag{5}$$

It remains to bound $\sum_{t=1}^{T} \mathrm{TV}_t$. By Pinsker's inequality, $\mathrm{TV}_t \leq \sqrt{\mathrm{KL}_t / 2}$ where $\mathrm{KL}_t \triangleq \mathrm{KL}(p_{\mathbf{v}}(\cdot \mid \mathbf{x}) \parallel q_{\mathbf{w}_t}(\cdot \mid \mathbf{x}))$. Applying the Cauchy-Schwarz inequality yields

$$\sum_{t=1}^{T} \mathrm{TV}_t \leq \sum_{t=1}^{T} \sqrt{\frac{\mathrm{KL}_t}{2}} \leq \sqrt{\frac{T}{2} \sum_{t=1}^{T} \mathrm{KL}_t} = \sqrt{\frac{T \cdot \mathbf{Reg}_T}{2}}, \tag{6}$$

where the last equality follows from Eq. (4) and the definition of dynamic regret in Eq. (2). Substituting Eq. (6) into Eq. (5), we arrive at

$$\mathbb{E}[|\hat{\mathbf{x}}|] \geq \frac{(k+1)\,T}{1 + (k+1)\sqrt{\mathbf{Reg}_T/(2T)}}.$$

On the other hand, the generation-refinement framework generates at most $k + 1$ tokens at each step $t$, and thus $\mathbb{E}[|\hat{\mathbf{x}}|] \leq (k+1) \cdot T$, which completes the proof. $\qquad\square$

### D.2. Proof of Theorem 1

*Proof.* By definition, the acceleration rate $\gamma$ is the speedup ratio of generation-refinement frameworks compared to standard autoregressive decoding. Without acceleration, generating $\mathbb{E}[|\hat{\mathbf{x}}|]$ tokens requires $A \cdot \mathbb{E}[|\hat{\mathbf{x}}|]$. With the generation-refinement framework, each of the $T$ steps involves: *(i)* drafting $k$ candidate tokens with the draft model, incurring cost $a \cdot k$, and *(ii)* parallel verification by the target model, incurring cost $A$. Thus, the total cost is $(a \cdot k + A) \cdot T$, yielding

$$\gamma = \frac{A \cdot \mathbb{E}[|\hat{\mathbf{x}}|]}{a \cdot k \cdot T + A \cdot T} = \frac{\mathbb{E}[|\hat{\mathbf{x}}|]}{T(\alpha k + 1)}, \tag{7}$$

where $\alpha = a/A$ denotes the inference time ratio.

**Upper bound.** By Lemma 1, we have $\mathbb{E}[|\hat{\mathbf{x}}|] \leq (k+1) \cdot T$. Substituting into (7) gives

$$\gamma \leq \frac{(k+1) \cdot T}{T(\alpha k + 1)} = \frac{k+1}{\alpha k + 1}. \tag{8}$$

**Lower bound.** By Lemma 1, we also have

$$\mathbb{E}[|\hat{\mathbf{x}}|] \geq \frac{(k+1)\,T}{1 + (k+1)\sqrt{\mathbf{Reg}_T/(2T)}}.$$

Substituting into (7) yields

$$\gamma \geq \frac{k+1}{(\alpha k + 1)\big(1 + (k+1)\sqrt{\mathbf{Reg}_T/(2T)}\big)}. \tag{9}$$

Therefore, combining (8) and (9), the acceleration rate satisfies

$$\frac{k+1}{(\alpha k + 1)\big(1 + (k+1)\sqrt{\mathbf{Reg}_T/(2T)}\big)} \leq \gamma \leq \frac{k+1}{\alpha k + 1},$$

which completes the proof. $\qquad\square$

### D.3. Proof of Corollary 1

*Proof.* We prove the dynamic regret bound for OGD and then derive the corresponding acceleration rate.

**Step 1: Dynamic regret bound.** We apply Theorem 1 in (Zhao et al., 2024). For OGD with a constant learning rate $\eta$, the per-round regret satisfies

$$f_t(\mathbf{w}_t) - f_t(\mathbf{w}_t^{\star}) \leq \frac{1}{2\eta}\left(\|\mathbf{w}_t^{\star} - \mathbf{w}_t\|_2^2 - \|\mathbf{w}_t^{\star} - \mathbf{w}_{t+1}\|_2^2\right) + \frac{\eta}{2}\|\nabla f_t(\mathbf{w}_t)\|_2^2.$$

Summing over $t = 1, \ldots, T$, we obtain

$$\mathbf{Reg}_T = \sum_{t=1}^{T} \left( f_t(\mathbf{w}_t) - f_t(\mathbf{w}_t^\star) \right) \leq \sum_{t=1}^{T} \frac{1}{2\eta} \left( \|\mathbf{w}_t^\star - \mathbf{w}_t\|_2^2 - \|\mathbf{w}_t^\star - \mathbf{w}_{t+1}\|_2^2 \right) + \sum_{t=1}^{T} \frac{\eta}{2} \|\nabla f_t(\mathbf{w}_t)\|_2^2. \qquad (10)$$

We now bound the first term on the right-hand side. By rearranging and adding intermediate terms, we have

$$\sum_{t=1}^{T} \left( \|\mathbf{w}_t^\star - \mathbf{w}_t\|_2^2 - \|\mathbf{w}_t^\star - \mathbf{w}_{t+1}\|_2^2 \right) = \|\mathbf{w}_1^\star - \mathbf{w}_1\|_2^2 - \|\mathbf{w}_T^\star - \mathbf{w}_{T+1}\|_2^2 + \sum_{t=2}^{T} \left( \|\mathbf{w}_t^\star - \mathbf{w}_t\|_2^2 - \|\mathbf{w}_{t-1}^\star - \mathbf{w}_t\|_2^2 \right).$$

The first two terms are bounded by the diameter of the constraint set. For the summation term, we use the identity $\|a\|^2 - \|b\|^2 = \langle a - b, a + b \rangle$ to obtain

$$\begin{aligned}
\|\mathbf{w}_t^\star - \mathbf{w}_t\|_2^2 - \|\mathbf{w}_{t-1}^\star - \mathbf{w}_t\|_2^2 &= \langle \mathbf{w}_t^\star - \mathbf{w}_{t-1}^\star, \mathbf{w}_t^\star + \mathbf{w}_{t-1}^\star - 2\mathbf{w}_t \rangle \\
&\leq \|\mathbf{w}_t^\star - \mathbf{w}_{t-1}^\star\|_2 \cdot \|\mathbf{w}_t^\star + \mathbf{w}_{t-1}^\star - 2\mathbf{w}_t\|_2 \\
&\leq 2D \|\mathbf{w}_t^\star - \mathbf{w}_{t-1}^\star\|_2,
\end{aligned}$$

where $D$ denotes the diameter of the constraint set $\mathcal{W}$. Summing over $t = 2, \ldots, T$, we get

$$\sum_{t=2}^{T} \left( \|\mathbf{w}_t^\star - \mathbf{w}_t\|_2^2 - \|\mathbf{w}_{t-1}^\star - \mathbf{w}_t\|_2^2 \right) \leq 2D \sum_{t=2}^{T} \|\mathbf{w}_t^\star - \mathbf{w}_{t-1}^\star\|_2 = 2DP_T,$$

where $P_T = \sum_{t=1}^{T} \|\mathbf{w}_{t+1}^\star - \mathbf{w}_t^\star\|_2$ is the path length of the comparator sequence.

Under Assumption 2, $\|\nabla f_t(\mathbf{w}_t)\|_2 \leq G$ for all $t$, substituting this into (10) yields

$$\mathbf{Reg}_T \leq \frac{D^2}{\eta} + \frac{DP_T}{\eta} + \frac{\eta G^2 T}{2}.$$

Setting $\eta = \mathcal{O}(1/\sqrt{T})$, specifically $\eta = \frac{D}{G\sqrt{T}}$, we obtain

$$\mathbf{Reg}_T \leq DG\sqrt{T} + DG\sqrt{T} \cdot P_T + \frac{DG\sqrt{T}}{2} = \mathcal{O}\left( \sqrt{T}(1 + P_T) \right).$$

**Step 2: Acceleration rate.** By Theorem 1, the acceleration rate satisfies

$$\gamma \geq \frac{k+1}{(\alpha k + 1)\left(1 + (k+1)\sqrt{\mathbf{Reg}_T/(2T)}\right)}.$$

Substituting the regret bound $\mathbf{Reg}_T = \mathcal{O}(\sqrt{T}(1 + P_T))$ into the above expression, we have

$$\sqrt{\frac{\mathbf{Reg}_T}{T}} = \mathcal{O}\left( \frac{\sqrt{1 + P_T}}{T^{1/4}} \right).$$

Since $\frac{a}{1+ab} \geq \frac{1}{2} \min\{a, 1/b\}$ for any $a, b > 0$, applying this with $a = k + 1$ and $b = \sqrt{\mathbf{Reg}_T/(2T)}$ yields

$$\gamma = \Omega\left( \frac{1}{\alpha k + 1} \min\left\{ k+1, \frac{T^{1/4}}{\sqrt{1 + P_T}} \right\} \right),$$

which completes the proof. □

**D.4. Proof of Corollary 2**

*Proof.* We prove the regret bound for optimistic online learning and the acceleration rate as follows.

**Step 1: Dynamic regret bound.** Recall that the two-step update in optimistic online learning in Eq. (3) is given by

$$\mathbf{w}_t = \Pi_{\mathcal{W}}\left[\widehat{\mathbf{w}}_t - \eta \mathbf{h}_t\right]; \quad \widehat{\mathbf{w}}_{t+1} = \Pi_{\mathcal{W}}\left[\widehat{\mathbf{w}}_t - \eta \nabla f_t(\mathbf{w}_t)\right],$$

where $\Pi_{\mathcal{W}}[\cdot]$ denotes the projection onto the domain $\mathcal{W}$, $\mathbf{h}_t$ is the hint (optimistic gradient), and $\nabla f_t(\mathbf{w}_t)$ is the true gradient. We consider the regularizer $\psi(\mathbf{w}) = \frac{1}{2}\|\mathbf{w}\|_2^2$, which is 1-strongly convex with respect to $\|\cdot\|_2$.

We again apply Theorem 1 in (Zhao et al., 2024). Summing over $t = 1, \ldots, T$, we have

$$\mathbf{Reg}_T = \sum_{t=1}^{T}\left(f_t(\mathbf{w}_t) - f_t(\mathbf{w}_t^\star)\right) \leq \eta \delta_T + \sum_{t=1}^{T}\frac{1}{2\eta}\left(\|\mathbf{w}_t^\star - \widehat{\mathbf{w}}_t\|_2^2 - \|\mathbf{w}_t^\star - \widehat{\mathbf{w}}_{t+1}\|_2^2\right). \tag{11}$$

Following the same telescoping argument as in the proof of Corollary 1, we have

$$\sum_{t=1}^{T}\left(\|\mathbf{w}_t^\star - \widehat{\mathbf{w}}_t\|_2^2 - \|\mathbf{w}_t^\star - \widehat{\mathbf{w}}_{t+1}\|_2^2\right) = \|\mathbf{w}_1^\star - \widehat{\mathbf{w}}_1\|_2^2 - \|\mathbf{w}_T^\star - \widehat{\mathbf{w}}_{T+1}\|_2^2 + \sum_{t=2}^{T}\left(\|\mathbf{w}_t^\star - \widehat{\mathbf{w}}_t\|_2^2 - \|\mathbf{w}_{t-1}^\star - \widehat{\mathbf{w}}_t\|_2^2\right).$$

Under Assumption 1, the first two terms are bounded by the diameter $D$ of the constraint set. For the summation term, using the identity $\|a\|^2 - \|b\|^2 = \langle a - b, a + b \rangle$ and the Cauchy-Schwarz inequality, we obtain $\|\mathbf{w}_t^\star - \widehat{\mathbf{w}}_t\|_2^2 - \|\mathbf{w}_{t-1}^\star - \widehat{\mathbf{w}}_t\|_2^2 \leq 2D\|\mathbf{w}_t^\star - \mathbf{w}_{t-1}^\star\|_2$. Summing over $t = 2, \ldots, T$, we get

$$\sum_{t=2}^{T}\left(\|\mathbf{w}_t^\star - \widehat{\mathbf{w}}_t\|_2^2 - \|\mathbf{w}_{t-1}^\star - \widehat{\mathbf{w}}_t\|_2^2\right) \leq 2DP_T,$$

where $P_T = \sum_{t=1}^{T}\|\mathbf{w}_{t+1}^\star - \mathbf{w}_t^\star\|_2$ is the path length. Substituting into (11), we obtain $\mathbf{Reg}_T \leq \eta \delta_T + \frac{D^2 + 2DP_T}{2\eta}$. Setting the learning rate as $\eta = \frac{D}{\sqrt{1+\delta_T}}$, we obtain

$$\mathbf{Reg}_T \leq \mathcal{O}\left(\sqrt{1+\delta_T} \cdot (1 + P_T)\right).$$

**Step 2: Acceleration rate.** By Theorem 1, the acceleration rate satisfies

$$\gamma \geq \frac{k+1}{(\alpha k + 1)\left(1 + (k+1)\sqrt{\mathbf{Reg}_T/(2T)}\right)}.$$

Substituting the regret bound $\mathbf{Reg}_T = \mathcal{O}\left(\sqrt{1+\delta_T} \cdot (1 + P_T)\right)$ into the above expression, we have

$$\sqrt{\frac{\mathbf{Reg}_T}{T}} = \mathcal{O}\left(\frac{(1+\delta_T)^{1/4}\sqrt{1+P_T}}{\sqrt{T}}\right).$$

Since $\frac{a}{1+ab} \geq \frac{1}{2}\min\{a, 1/b\}$ for any $a, b > 0$, applying this with $a = k+1$ and $b = \sqrt{\mathbf{Reg}_T/(2T)}$ yields

$$\gamma = \Omega\left(\frac{1}{\alpha k + 1}\min\left\{k+1, \frac{\sqrt{T}}{(1+\delta_T)^{1/4}\sqrt{1+P_T}}\right\}\right),$$

which completes the proof. □

**D.5. Proof of Corollary 3**

*Proof.* We prove the dynamic regret bound for the online ensemble learning and derive the corresponding acceleration rate.

**Step 1: Dynamic regret bound.** Following the online ensemble framework (Zhao et al., 2024), we maintain $N$ base learners with geometrically spaced learning rates:

$$\eta_i = \frac{2^{i-1}D}{G}\sqrt{\frac{1}{T}}, \quad i = 1, \ldots, N,$$

where $D$ denotes the diameter of the constraint set $\mathcal{W}$, $G$ is the gradient bound, and the number of base learners is set to $N = \left\lceil \frac{1}{2}\log_2(1+T)\right\rceil + 1 = \mathcal{O}(\log T)$. Each base learner $\mathbf{w}_t^i$ is updated via OGD with learning rate $\eta_i$:

$$\mathbf{w}_{t+1}^i = \Pi_{\mathcal{W}}\left[\mathbf{w}_t^i - \eta_i \nabla f_t(\mathbf{w}_t^i)\right].$$

The meta learner combines the outputs of base learners using the Hedge algorithm with probability weights $p_t^i \propto \exp\left(-\varepsilon\sum_{s=1}^{t-1} f_s(\mathbf{w}_s^i)\right)$, where $\varepsilon > 0$ is the step size. The final output is $\mathbf{w}_t = \sum_{i=1}^{N} p_t^i \cdot \mathbf{w}_t^i$. Following Zhao et al. (2024), the geometric spacing of learning rates ensures that at least one base learner achieves near-optimal performance for any path length $P_T$. Specifically, there exists an index $i^\star$ such that the base learner with learning rate $\eta_{i^\star}$ achieves

$$\min_{i\in[N]}\sum_{t=1}^{T}\left(f_t(\mathbf{w}_t^i) - f_t(\mathbf{w}_t^\star)\right) \leq \mathcal{O}\left(\sqrt{T(1+P_T)}\right).$$

Further apply Theorem 1 in (Zhao et al., 2024), the overall dynamic regret of the online ensemble is

$$\mathbf{Reg}_T \leq \mathcal{O}\left(\sqrt{T(1+P_T)} + \sqrt{T\log\log T}\right) = \mathcal{O}\left(\sqrt{T(1+P_T)}\right).$$

**Step 2: Acceleration rate.** By Theorem 1, the acceleration rate satisfies

$$\gamma \geq \frac{k+1}{(\alpha k+1)\left(1+(k+1)\sqrt{\mathbf{Reg}_T/(2T)}\right)}.$$

Substituting the regret bound $\mathbf{Reg}_T = \mathcal{O}(\sqrt{T(1+P_T)})$ into the above expression, we have

$$\sqrt{\frac{\mathbf{Reg}_T}{T}} = \mathcal{O}\left(\frac{(1+P_T)^{1/4}}{T^{1/4}}\right).$$

Since $\frac{a}{1+ab} \geq \frac{1}{2}\min\{a, 1/b\}$ for any $a, b > 0$, applying this with $a = k+1$ and $b = \sqrt{\mathbf{Reg}_T/(2T)}$ yields

$$\gamma = \Omega\left(\frac{1}{\alpha k+1}\min\left\{k+1, \frac{T^{1/4}}{(1+P_T)^{1/4}}\right\}\right),$$

which completes the proof. $\qquad\square$

# E. Limitation

Our experiments train initial draft models from scratch on offline data, rather than directly adopting publicly released checkpoints. This design enables a controlled evaluation of whether online adaptation can continuously improve draft quality under diverse user inputs starting from the same offline dataset. Overall, we emphasize that our primary contribution lies in the OnlineSPEC framework: establishing a formal connection between speculative decoding and online learning, and enabling systematic algorithm design via the rich online learning toolkit, rather than benchmark-specific performance gains.

