# OpenReview forum: "When Drafts Evolve: Speculative Decoding Meets Online Learning"
_ICML.cc/2026/Conference — ICML 2026 regular_

### Official Review · Reviewer_rSxG · 2026-03-04

**Soundness:** 2
**Presentation:** 3
**Significance:** 3
**Originality:** 2
**Overall Recommendation:** 4
**Confidence:** 4

**Summary:**

This paper proposes OnlineSPEC, a unified framework that formulates speculative decoding as an online learning process. The key idea is to leverage the verification feedback from the target model to continuously adapt the draft model during inference, forming an iterative loop of draft generation, verification, and model update. The authors establish a theoretical connection between the dynamic regret of the online learning algorithm and the acceleration rate of speculative decoding, providing a theoretical perspective on how draft model adaptation can improve acceptance rate and inference efficiency. Based on this framework, the paper presents three instantiations—Online-LR, Opt-Hydra, and Ens-Eagle—which incorporate different online learning techniques such as online gradient descent, optimistic learning, and ensemble learning. Experiments are conducted on multiple benchmarks covering reasoning, code generation, and question answering tasks, using several language models. The results show that the proposed methods can improve inference speed while maintaining comparable output quality.

**Compliance With Llm Reviewing Policy:**

Affirmed.

**Final Justification:**

done

**Key Questions For Authors:**

**1.** What is the rationale behind pairing specific online learning strategies with particular speculative decoding methods (e.g., optimistic learning with Hydra and ensemble learning with EAGLE)? Are there structural reasons for these choices, or are they primarily empirical?

**2.** How sensitive is the performance of the proposed methods to the number of offline warm-up samples? For example, how would the results change if the warm-up dataset size were increased or decreased (e.g., 500, 2000 samples)?

**3.** Have the authors evaluated the proposed framework on larger models (e.g., 30B or 70B)? If not, do the authors anticipate any scalability challenges when applying OnlineSPEC to larger LLMs?

**Limitations:**

**yes**

**Strengths And Weaknesses:**

### **Strengths:**

**1. Novel perspective connecting speculative decoding and online learning.**

The paper proposes the OnlineSPEC framework, which formulates the interaction between the draft model and the target model in speculative decoding as an online learning process. This perspective introduces an interesting viewpoint and provides a useful framework for understanding how verification feedback can be leveraged to improve inference acceleration.

**2. Theoretical analysis relating dynamic regret to acceleration rate.**

The paper establishes a connection between dynamic regret in online learning and the acceleration rate of speculative decoding. This analysis offers a theoretical perspective on how continuously adapting the draft model using verification feedback may improve the acceptance rate and inference efficiency.

**3. Empirical validation across multiple tasks and models.**

The paper presents three instantiations of the framework (Online-LR, Opt-Hydra, and Ens-Eagle) and evaluates them on several tasks and datasets, including mathematical reasoning, code generation, and question answering. Experiments on multiple language models show that the proposed approach can improve inference speed while maintaining output quality.

**4. Clear presentation and organization.**

The paper is generally well written and clearly organized. The motivation is clearly explained, and the methodology and theoretical analysis are presented in a coherent manner. The figures (e.g., the framework illustration) further help clarify the proposed approach.

### **Weaknesses:**

 **1. Limited algorithmic novelty.**

Although the paper proposes the OnlineSPEC framework, the instantiated algorithms largely rely on existing online learning techniques, such as Online Gradient Descent, optimistic online learning, and ensemble learning. As a result, the contribution mainly lies in the framework perspective and its application to speculative decoding, rather than introducing substantially new algorithms.

**2. Insufficient justification for method combinations.**

The paper presents three instantiations (Online-LR, Opt-Hydra, and Ens-Eagle), but the rationale behind these specific combinations is not fully explained. For example, Hydra could also be combined with ensemble learning, while EAGLE might be integrated with optimistic updates. The paper does not provide clear design principles or theoretical reasoning explaining why particular online learning strategies are paired with certain speculative decoding methods. In addition, ablation studies comparing alternative combinations are missing, making the current design choices appear somewhat empirical.

**3. Lack of analysis on the offline warm-up dataset size.**

The experimental setup uses 1000 samples for the offline warm-up stage followed by 4000 samples in the online phase, but the impact of the warm-up dataset size is not analyzed. For methods such as EAGLE or EAGLE-3, the quality of the draft head may be sensitive to the amount of training data. Without studying the effect of different warm-up sizes, it is difficult to determine whether the observed improvements mainly stem from the proposed online learning framework or from the initialization quality of the draft model.

**4. Limited experimental scale.**

Experiments are conducted on mid-scale models (around 7B–8B parameters). Since speculative decoding is typically deployed with much larger models (e.g., 30B–70B), it remains unclear whether the proposed approach would maintain similar effectiveness and stability at larger scales.

---

> ### Author Rebuttal · Authors · 2026-03-30
>
> Thanks for your helpful comments. Below, we address your technical questions and resolve any potential misunderstandings.
>
> ---
>
> **Q1.** "What is the rationale behind pairing specific online learning strategies with particular speculative decoding methods?"
>
> **A1.** We thank the reviewer for this question. The combination of specific online learning strategies with particular speculative decoding methods is motivated by their *structural compatibility*. Specifically, Hydra employs sequentially dependent draft heads that exploit contextual dependencies across positions, which naturally aligns with optimistic online learning that reuses historical gradients as predictive hints. Similarly, EAGLE constructs tree-structured draft sequences with multiple candidate branches, which naturally corresponds to ensemble learning that maintains and combines diverse base learners. We further validate these design choices empirically on GSM8K with Vicuna-13B:
>
> |Method|AVGLEN|SPEEDUP|
> |-|:-:|:-:|
> |Hydra (baseline)|2.22|1.00× (TPS 60.61)|
> |OSD-Hydra|2.50|1.13× (TPS 68.36)|
> |Ens-Hydra (ensemble)|2.61|1.15× (TPS 70.64)|
> |**Opt-Hydra (optimistic)**|**2.73**|**1.23× (TPS 74.37)**|
>
> The results demonstrate that although both online learning strategies improve over offline baselines, the proposed pairings consistently achieve the best performance for each speculative decoding method. This supports that our design choices are grounded in their *structural compatibility*. We will incorporate this analysis into the next version.
>
> ---
>
> **Q2.** "How sensitive is the performance of the proposed methods to the number of offline warm-up samples?"
>
> **A2.** We thank the reviewer for the comment. We conduct additional experiments to analyze the sensitivity to warm-up dataset size on GSM8K with Vicuna-7B:
>
> |Warm-up Samples|AVGLEN OSD-EAGLE|AVGLEN Ens-EAGLE|TPS OSD-EAGLE|TPS Ens-EAGLE|
> |-|:-:|:-:|:-:|:-:|
> |500|1.75|1.78|63.28|65.52|
> |1000|1.88|1.91|69.66|70.82|
> |1500|2.03|2.17|76.13|80.07|
>
> The results show that across all warm-up sizes, Ens-EAGLE consistently outperforms OSD-EAGLE. Notably, Ens-EAGLE has better improvements with larger warm-up sizes (e.g., 1500 samples), as stronger initialization yields better-trained and more diverse base learners, allowing ensemble learning to achieve better performance by adaptively combining them. These results demonstrate the robustness of our method across different warm-up dataset sizes. We will include this analysis in the revised version.
>
> ---
>
> **Q3.** "Have the authors evaluated the proposed framework on larger models (e.g., 30B or 70B)?"
>
> **A3.** We thank the reviewer for this suggestion. We have conducted additional experiments on larger-scale models to evaluate the scalability of our framework. Specifically, we evaluate Vicuna-13B and Qwen3-32B on GSM8K dataset, with results reported below:
>
> - **EAGLE-3: Vicuna-13B-v1.3 on GSM8K**
>
>   |Method|SPEEDUP ↑|AVGLEN ↑|
>   |-|:-:|:-:|
>   |Standard AR|1.00 (40.40 TPS)|1.00|
>   |Vanilla SD|1.20 (48.52 TPS)|1.93|
>   |OSD-EAGLE-3|1.37 (55.52 TPS)|2.24|
>   |**Ens-EAGLE-3**|**1.46 (59.03 TPS)**|**2.35**|
>
> - **Hydra: Vicuna-13B-v1.3 on GSM8K**
>
>   |Method|SPEEDUP ↑|AVGLEN ↑|
>   |-|:-:|:-:|
>   |Standard AR|1.00 (40.40 TPS)|1.00|
>   |Vanilla SD|1.50 (60.61 TPS)|2.22|
>   |OSD-Hydra|1.69 (68.36 TPS)|2.50|
>   |**Opt-Hydra**|**1.84 (74.37 TPS)**|**2.73**|
>
> - **Online-LR: Qwen3-32B + Qwen3-1.7B on GSM8K**
>
>   |Method|SPEEDUP ↑|AVGLEN ↑|ACC (%) ↑|
>   |-|:-:|:-:|:-:|
>   |Standard AR|1.00 (35.53 TPS)|1.00|96.08|
>   |LR|1.11 (39.39 TPS)|3.66|95.76|
>   |OSD-LR|1.09 (38.63 TPS)|3.18|95.44|
>   |**Online-LR**|**1.31 (46.71 TPS)**|**9.98**|95.68|
>
>
> The results demonstrate that OnlineSPEC consistently improves over both baselines across larger model scales, confirming the scalability of our framework. We will add these results to the revised manuscript. Thank you!
>
> ---
>
> **Q4.** Regarding the novelty.
>
> **A4.** Thanks for the comment. We would like to clarify that the primary contribution of OnlineSPEC lies in it providing a unified framework, rather than ad-hoc algorithmic modifications. Specifically, our work is the first to establish a formal connection between speculative decoding and online learning, and to theoretically link dynamic regret to the acceleration rate (Theorem 1). This connection enables systematically incorporating advanced online learning techniques into speculative decoding. Moreover, OnlineSPEC can be seamlessly integrated with existing speculative decoding methods to further improve their performance. We hope this addresses the reviewer's concern.
>
> ---
>
> **In summary, we have:**
>
> - Provided a detailed analysis of method combination, along with ablation experiments validating these design choices.
> - Conducted additional experiments on warm-up data sensitivity and larger model scales, following your suggestion.
>
> We sincerely appreciate your helpful feedback. If our responses have addressed your concerns, we would be grateful if you would consider raising the score. Thank you!

---

> > ### Author Rebuttal · Reviewer_rSxG · 2026-04-01
> >
> > Thank you for the additional experiments. Your rebuttal has addressed my main concerns regarding method pairing rationale (Q1), warm-up sensitivity (Q2), and scalability to larger models (Q3). The ablation studies on Hydra and the experiments on Vicuna-13B and Qwen3-32B provide convincing empirical validation. I acknowledge that the primary contribution is the unified framework connecting speculative decoding and online learning, along with the theoretical analysis.
> > I am raising my score from 3 (Weak Reject) to 4 (Borderline Accept). Please incorporate these additional results into the final manuscript.
> > Updated Recommendation: 4 (Borderline Accept)

---

> > > ### Author Response · Authors · 2026-04-07
> > >
> > > Dear Reviewer rSxG,
> > >
> > > We sincerely thank the reviewer for the constructive feedback and for raising the score. We will carefully incorporate these additional experimental results into the revised manuscript and improve the presentation accordingly.
> > >
> > > We would also like to kindly remind that the *recommendation score has not yet been updated*, we would greatly appreciate it if you could consider **updating the final justification and also the overall score** in the openreview system. Thank you very much for your support and efforts!
> > >
> > > Best regards,
> > >
> > > Authors

---

### Official Review · Reviewer_inDi · 2026-03-12

**Soundness:** 3
**Presentation:** 3
**Significance:** 4
**Originality:** 4
**Overall Recommendation:** 5
**Confidence:** 3

**Summary:**

The paper observes that the feedback generated during speculative decoding can be leveraged to update the draft model, and formulates this process as an online learning problem. Building on this insight, the authors propose a unified framework that systematically utilizes interactive feedback to continuously improve the draft model during inference. The paper further establishes a formal connection between online learning performance and the acceleration rate of speculative decoding systems. Based on this framework, the authors develop three algorithms, Online-LR, Opt-Hydra, and Ens-Eagle, by leveraging modern online learning techniques such as optimistic online learning and online ensemble learning.

**Compliance With Llm Reviewing Policy:**

Affirmed.

**Final Justification:**

The authors acknowledge the limitations of the i.i.d. assumption and provide a reasonable justification aligned with their short-horizon empirical setting. They also support generalization and efficiency claims with cross-domain experiments and concrete cost–performance analysis. Finally, since the work is positioned as a unifying framework, the relatively lighter discussion of prior methods is reasonable and does not undermine the contribution.

**Key Questions For Authors:**

## Key Questions

1. **Justification of the i.i.d. assumption.**
   The theoretical analysis assumes that the feedback signals satisfy an i.i.d. assumption, following prior work. Could the authors clarify why the i.i.d. assumption is reasonable in this setting, or discuss how sensitive the theoretical guarantees are if this assumption does not strictly hold?

2. **Generalization of the draft model across benchmarks.**
   The draft model is continuously updated through online learning during decoding. How well does the updated draft model generalize across different tasks or benchmarks? For example, does the draft model trained or adapted on one dataset retain its performance on others, or does it require task-specific adaptation?

**Limitations:**

yes

**Strengths And Weaknesses:**

## 1. Soundness

### Strengths
- The paper’s motivation is natural and the perspective is novel. Previous work on speculative decoding typically treats draft model updates as heuristic procedures, whereas this paper explicitly formulates the process as an online learning problem.
- The paper clearly highlights the conceptual similarity between speculative decoding and online learning.
- Based on this perspective, the proposed algorithms are supported by theoretical acceleration-rate guarantees derived from regret analysis.
- The experiments are well designed and evaluate the approach across multiple benchmarks. The empirical results consistently support the claims and demonstrate pronounced improvements.

### Weaknesses
1. Assumption 1 is adopted from prior work and appears reasonable, but it is not entirely clear why the i.i.d. assumption should hold in this setting. Additional explanation or justification would be helpful.
2. The computational costs of the three proposed algorithms are not discussed. While the draft model may be relatively small, it would still be useful to provide a brief summary of the computational overhead or the runtime complexity of the algorithms.

---

## 2. Presentation

### Strengths
- The paper is clearly written and well structured.
- The overall narrative is easy to follow.
- The related work section effectively positions the paper within the existing literature.

### Weaknesses
- The paper could place more emphasis on the motivation for refining previous methods and more clearly articulate the differences between the proposed framework and prior approaches.

---

## 3. Significance

### Strengths
- The paper addresses a practically important problem in accelerating LLM inference.
- The proposed framework and algorithms are conceptually simple and could be adopted by practitioners.
- The empirical gains in accuracy are meaningful.

### Weaknesses
1. As mentioned earlier, the trade-off between computational cost and performance is not thoroughly analyzed.
2. The theoretical results provide guarantees on performance through regret bounds. It would be helpful to provide concrete numerical examples of these bounds under typical parameter settings to better illustrate their practical implications.

---

## 4. Originality

### Strengths
- The paper introduces a new perspective on speculative decoding by connecting it with the well-established literature on online learning.
- The framework integrates several modern online learning techniques into the speculative decoding setting.
- The contributions are clearly distinguished from closely related work.

---

> ### Author Rebuttal · Authors · 2026-03-30
>
> Thanks for your helpful comments! Below, we address your technical concerns.
>
> ---
>
> **Q1.** "Justification of the i.i.d. assumption."
>
> **A1.** Thank you for this question. Following prior work (Leviathan et al., 2023), we adopt the i.i.d. assumption as a simplified abstraction to study the theoretical properties of speculative decoding. We acknowledge that this is arguably a strong and restrictive assumption for token generation. In our view, it could be relaxed to weaker dependence structures (such as Markov or more general mixing processes), which would be an interesting direction for future work.
>
> On the other hand, in our experiments, we set the draft model to generate only k=4-8 tokens per step. Over such a short window, distribution shifts across positions are mitigated, making the i.i.d. assumption a reasonable approximation. This design choice aligns the theory more closely with our empirical design. Indeed, our experiments across seven benchmarks demonstrate consistent improvements, indicating the robustness of our framework in practice. We will add this discussion to the revised manuscript. Thank you!
>
> ---
>
> **Q2.** "Generalization of the draft model across benchmarks."
>
> **A2.** Thank you for this insightful question. We conducted a cross-domain evaluation to assess the generalization capability of the updated draft model. Specifically, we trained Ens-EAGLE-3 with Vicuna-13B-v1.3 on the Code-Search-Python dataset, then directly evaluated it on the Spider dataset. The results are shown below:
>
> | Training Data | Speedup | AvgLen |
> |-|:-:|:-:|
> | Code-Search-Python | 1.13× (45.81 TPS) | 1.89 |
> | Spider  | 1.48× (59.69 TPS) | 2.48 |
>
> The results demonstrate that the draft model adapted on Code-Search-Python retains reasonable acceleration capability on Spider (1.13× speedup), indicating some degree of cross-domain generalization. However, the domain-specific adapted model (trained on Spider) achieves better performance (1.48× speedup), suggesting that task-specific adaptation is beneficial for optimal performance. This aligns with our theoretical framework: online learning enables the draft model to adapt to the target distribution, and domain-specific data provides more relevant feedback signals. We will include this analysis in the revised manuscript. Thank you!
>
> ---
>
> **Q3.** "The trade-off between computational cost and performance is not thoroughly analyzed... provide concrete numerical examples of these bounds under typical parameter settings to better illustrate their practical implications."
>
> **A3.** Thank you for this valuable feedback. We provide a detailed analysis of the computational complexity and theoretical acceleration guarantees for each algorithm below:
>
> | Algorithm | Per-Step Update Complexity | Lower Bound of Acceleration Rate|
> |-|:-:|:-:|
> | **Online-LR** | $\mathcal{O}( \|S_t\| \cdot d)$ | $\tilde{\Omega}\left(\frac{(1-1/k) \cdot T^{1/4}}{(\alpha k+1)\sqrt{1+P_T}}\right)$ |
> | **Opt-Hydra** | $\mathcal{O}(2 \|S_t\| \cdot d)$ | $\tilde{\Omega}\left(\frac{(1-1/k) \cdot \sqrt{T}}{(\alpha k+1)(1+\delta_T)^{1/4}\sqrt{1+P_T}}\right)$ |
> | **Ens-EAGLE** | $\mathcal{O}(N \cdot  \|S_t\| \cdot d)$ | $\tilde{\Omega}\left(\frac{(1-1/k) \cdot T^{1/4}}{(\alpha k+1)(1+P_T)^{1/4}}\right)$ |
>
> Here, $|S_t|$ denotes the feedback batch size, $d$ is the draft model parameter count, $N$ is the number of base learners, $P_T$ is the path length, and $\delta_T$ measures hint quality. Besides, we further provide visualizations comparing theoretical bounds with observed speedup (https://anonymous.4open.science/r/OnlineSPEC/rebuttal/rebuttal.md), which turns out to have the same trends. We will add this analysis to the revised manuscript. Thank you!
>
> ---
>
> **Q4.** "The paper could place more emphasis on the motivation for refining previous methods and more clearly articulate the differences between the proposed framework and prior approaches."
>
> **A4.** Thank you for this feedback. We clarify that OnlineSPEC's primary contribution is providing a **unified framework** rather than individual algorithmic improvements. Prior online methods (OSD, ATLAS) have achieved notable empirical success through task-specific designs. In contrast, OnlineSPEC offers a general formulation that systematically connects speculative decoding with online learning (Theorem 1). This unified perspective enables incorporating diverse online learning techniques (optimistic learning, ensemble learning) and can be **seamlessly integrated** with existing methods. We will emphasize this distinction more clearly in the revised manuscript. Thank you!

---

> > ### Author Rebuttal · Reviewer_inDi · 2026-04-02
> >
> > Thank you for the detailed rebuttal and additional experiments. The added analysis strengthens the connection between the theoretical framework and empirical results, and I find the consistency between the theoretical bounds and observed speedups convincing.
> > Although the i.i.d. assumption remains a relatively strong simplification, the empirical results support its practical validity under the proposed setup. The cross-domain results also suggest reasonable, though not perfect, generalization of the draft model, which is acceptable given the scope of the work.

---

> > > ### Author Response · Authors · 2026-04-04
> > >
> > > We appreciate the reviewer's insightful comments and positive evaluation of our contribution. Your constructive feedback on the i.i.d. assumption and cross-domain generalization is valuable, and we will refine the manuscript accordingly. Thank you!

---

### Official Review · Reviewer_4rmk · 2026-03-13

**Soundness:** 4
**Presentation:** 3
**Significance:** 3
**Originality:** 3
**Overall Recommendation:** 5
**Confidence:** 3

**Summary:**

This paper addresses speculative decoding in the context of streaming. More concretely, the authors propose that rather than learning a static draft model, one may adapt the model during subsequent inference steps. This can lead to longer acceptance lengths and higher token throughput for large language models (LLMs). The proposed method is grounded in online learning theory with proofs to support the theoretical claims and empirical evidence to illustrate the applicability of the method in real world use cases.

**Compliance With Llm Reviewing Policy:**

Affirmed.

**Final Justification:**

This is a technically sound, well presented paper with at least a moderate degree of impact. I support accepting the paper.

**Key Questions For Authors:**

It is not always clear to me what the draft model in the implementation details section. Can the authors clarify what it is?

**Limitations:**

Discussed in appendix E.

**Strengths And Weaknesses:**

**Soundness**: The paper is technically sound, the theoretical arguments are well supported via proofs in the appendix. Furthermore, the experiments provide sufficient evidence that the method works in practice. Some of the LLMs used are a little old (Vicuna and Llama 2) but Qwen 3 is reasonably modern. It would have been nice to use larger LLMs to verify if the experiments or efficiency gains are even larger.

**Presentation**: The paper is clear, well written and easy to follow. The figures have captions that are easy to understand.

**Significance**: The results of the paper are of moderate significance. I believe that the method can improve over existing speculative decoding methods, however, the results are not ground breaking as the performance increase is modest. Nevertheless, it appears to be consistent across enough settings.

**Originality**: While the paper hinges on insights from previous works and builds on top of methods such as OSD (Liu et al., 2024b), Hydra (Ankner et al., 2024), and EAGLE-3 (Li et al., 2025), it is still original. This is demonstrated by comparing to naive combinations of these methods in the appendix ablations and the theoretical work which is original and does not depend on these works.

---

> ### Author Rebuttal · Authors · 2026-03-30
>
> Thanks for your constructive and helpful comments! We provide our response to each question as below.
>
> ---
>
> **Q1.** "It is not always clear to me what the draft model in the implementation details section. Can the authors clarify what it is?"
>
> **A1.** Thank you for pointing this out. We clarify the draft model architecture for each method below:
>
> - **Hydra**: The draft model is a *HydraPrefixMLP* consisting of: (i) a prefix embedding layer implemented as one complete LlamaDecoder layer, (ii) four grounded MLP heads with progressively increasing input dimensions to capture sequential dependencies, and (iii) four language modeling heads for token prediction. The total parameter count is approximately 1.53B.
>
> - **EAGLE**: The draft model is a lightweight autoregressive model comprising a single Transformer layer with input and output projection layers, totaling approximately 0.24B parameters.
>
> - **EAGLE-3**: Similar to EAGLE, the draft model consists of a single Transformer layer with input and output layers, but with an increased capacity of approximately 0.5B parameters to improve prediction accuracy.
>
> - **Lookahead Reasoning (LR)**: We use Qwen3-0.6B as the draft model, which is a compact language model with 0.6B parameters.
>
> All draft models are initialized through offline training on domain-specific data using knowledge distillation from the target model, following the common practice in speculative decoding literature. We will add these details to the revised manuscript for clarity.
>
> ---
>
> **Q2.** "It would have been nice to use larger LLMs to verify if the experiments or efficiency gains are even larger."
>
> **A2.** We sincerely appreciate the reviewer for this question. We have conducted additional experiments on larger-scale models to evaluate the scalability of our framework. Specifically, we evaluate Vicuna-13B and Qwen3-32B on GSM8K dataset, with results reported below:
>
> - **EAGLE-3: Vicuna-13B-v1.3 on GSM8K**
>
>   |Method|SPEEDUP ↑|AVGLEN ↑|
>   |-|:-:|:-:|
>   |Standard AR|1.00 (40.40 TPS)|1.00|
>   |Vanilla SD|1.20 (48.52 TPS)|1.93|
>   |OSD-EAGLE-3|1.37 (55.52 TPS)|2.24|
>   |**Ens-EAGLE-3**|**1.46 (59.03 TPS)**|**2.35**|
>
> - **Hydra: Vicuna-13B-v1.3 on GSM8K**
>
>   |Method|SPEEDUP ↑|AVGLEN ↑|
>   |-|:-:|:-:|
>   |Standard AR|1.00 (40.40 TPS)|1.00|
>   |Vanilla SD|1.50 (60.61 TPS)|2.22|
>   |OSD-Hydra|1.69 (68.36 TPS)|2.50|
>   |**Opt-Hydra**|**1.84 (74.37 TPS)**|**2.73**|
>
> - **Online-LR: Qwen3-32B + Qwen3-1.7B on GSM8K**
>
>   |Method|SPEEDUP ↑|AVGLEN ↑|ACC (%) ↑|
>   |-|:-:|:-:|:-:|
>   |Standard AR|1.00 (35.53 TPS)|1.00|96.08|
>   |LR|1.11 (39.39 TPS)|3.66|95.76|
>   |OSD-LR|1.09 (38.63 TPS)|3.18|95.44|
>   |**Online-LR**|**1.31 (46.71 TPS)**|**9.98**|95.68|
>
> These results demonstrate that OnlineSPEC consistently improves over both baselines across larger model scales (**13B and 32B**), confirming the scalability of our framework. We will add these results to the revised manuscript. Thank you for the suggestion!

---

> > ### Author Rebuttal · Reviewer_4rmk · 2026-04-02
> >
> > Thank you for the response, I was already in favor of accepting the paper and will maintain my score.

---

> > > ### Author Response · Authors · 2026-04-04
> > >
> > > We sincerely thank the reviewer for the constructive feedback and positive evaluation of our work. We will carefully incorporate your valuable suggestions into the revised manuscript. Thank you!

---

### Official Review · Reviewer_LKVx · 2026-03-19

**Soundness:** 3
**Presentation:** 3
**Significance:** 4
**Originality:** 3
**Overall Recommendation:** 4
**Confidence:** 4

**Summary:**

This paper introduces OnlineSPEC, a framework that reformulates generation-refinement methods (like speculative decoding) as an online learning problem. The authors instantiate this framework through three specific applications: Online-LR for reasoning, Opt-Hydra with optimistic updates, and Ens-EAGLE using online ensemble learning. The work is notable for providing a unifying theoretical lens across seven benchmarks and three foundation models, reporting consistent speedup gains over static offline baselines while accounting for training-plus-inference overhead.

**Compliance With Llm Reviewing Policy:**

Affirmed.

**Final Justification:**

Thanks for your response, I'm willing to keep my score.

**Key Questions For Authors:**

1.  To better assess the significance of the 28% gain (OnlineSPEC vs. Offline Speculative), the authors should provide a direct comparison against standard AR decoding.

2. could the authors provide a visualization (e.g., a line chart or shaded area plot) comparing the Theoretical acceleration bounds derived in the paper against the actual observed speedup?

**Limitations:**

1. The theory-to-practice gap is substantial. It will be better to add details discussion on this gap.
2. Experiments with larger target models (eg, 13B and 33B) are essential to further improve the paper quality

**Strengths And Weaknesses:**

### Pros

1. The topic this paper focus on is useful and important to adapt draft models during inference.
2. The framework is flexible enough to support very different feedback types, including DPO-style reasoning feedback rather than only token-level speculative-decoding errors.
2. The paper is well-written and easy to follow. Anonymous code is provided.

### Cons

see Questions and Limitations

---

> ### Author Rebuttal · Authors · 2026-03-30
>
> We sincerely appreciate the reviewer's constructive feedback. In the following, we address each of your technical inquiries.
>
> ---
>
> **Q1.** "...the authors should provide a direct comparison against standard AR decoding."
>
> **A1.** Thank you for this suggestion. We have added a direct comparison against standard autoregressive (AR) decoding. The results on the GSM8K dataset are presented below, where we report tokens per second (TPS):
>
> | Method | Vicuna-7B-v1.3 | Llama-2-7B-Chat | Vicuna-13B-v1.3 |
> |-|:-:|:-:|:-:|
> | Standard AR | 54.11 | 53.76 | 40.40 |
> | Ens-EAGLE-3 | 73.96 (+36.7%) | 79.30 (+47.5%) | 59.03 (+46.1%) |
> | Opt-Hydra | 107.17 (+98.1%) | 122.61 (+128.1%) | 74.37 (+84.1%) |
>
> As shown, our methods achieve significant speedup over standard AR decoding. These results demonstrate that our OnlineSPEC framework substantially improves inference efficiency. We will include this analysis in the revised version.
>
> ---
>
> **Q2.** "Could the authors provide a visualization (e.g., a line chart or shaded area plot) comparing the Theoretical acceleration bounds derived in the paper against the actual observed speedup?"
>
> **A2.** Thank you for this suggestion. We note that directly visualizing theoretical bounds against empirical speedup presents several *challenges*: exact computation of quantities such as path length $P_T$ requires knowledge of the optimal time-varying comparators, which are unavailable in practice, and the bounds involve constant terms that make the comparison less straightforward.
>
> Nevertheless, our theory serves well to justify the algorithmic designs. Taking Ens-EAGLE as an example, we approximate $P_T$ by measuring the cumulative $\ell_2$-distance between consecutive prompts' embeddings as a proxy for path length. The **visualization** at https://anonymous.4open.science/r/OnlineSPEC/rebuttal/rebuttal.md shows that the theoretical trend aligns well with the observed speedup. We will incorporate this discussion into the revision.
>
> ---
>
> **Q3.** "Experiments with larger target models (eg, 13B and 33B) are essential to further improve the paper quality."
>
> **A3.** We sincerely appreciate the reviewer for this question. We have conducted additional experiments on larger-scale models to evaluate the scalability of our framework. Specifically, we evaluate Vicuna-13B and Qwen3-32B on GSM8K dataset, with results reported below:
>
> - **EAGLE-3: Vicuna-13B-v1.3 on GSM8K**
>
>   |Method|SPEEDUP ↑|AVGLEN ↑|
>   |-|:-:|:-:|
>   |Standard AR|1.00 (40.40 TPS)|1.00|
>   |Vanilla SD|1.20 (48.52 TPS)|1.93|
>   |OSD-EAGLE-3|1.37 (55.52 TPS)|2.24|
>   |**Ens-EAGLE-3**|**1.46 (59.03 TPS)**|**2.35**|
>
> - **Hydra: Vicuna-13B-v1.3 on GSM8K**
>
>   |Method|SPEEDUP ↑|AVGLEN ↑|
>   |-|:-:|:-:|
>   |Standard AR|1.00 (40.40 TPS)|1.00|
>   |Vanilla SD|1.50 (60.61 TPS)|2.22|
>   |OSD-Hydra|1.69 (68.36 TPS)|2.50|
>   |**Opt-Hydra**|**1.84 (74.37 TPS)**|**2.73**|
>
> - **Online-LR: Qwen3-32B + Qwen3-1.7B on GSM8K**
>
>   |Method|SPEEDUP ↑|AVGLEN ↑|ACC (%) ↑|
>   |-|:-:|:-:|:-:|
>   |Standard AR|1.00 (35.53 TPS)|1.00|96.08|
>   |LR|1.11 (39.39 TPS)|3.66|95.76|
>   |OSD-LR|1.09 (38.63 TPS)|3.18|95.44|
>   |**Online-LR**|**1.31 (46.71 TPS)**|**9.98**|95.68|
>
>
> The results demonstrate that OnlineSPEC consistently improves over both baselines across larger model scales (**13B** and **32B**), confirming the scalability of our framework. We will add these results to the revised manuscript. Thank you for the suggestion!

---

> > ### Author Rebuttal · Reviewer_LKVx · 2026-04-03
> >
> > Thanks for your response, I'm willing to keep my score.

---

### Decision · Program_Chairs · 2026-04-30

**Decision:**

Accept (regular)

**Comment:**

All reviewers are in favor of acceptance of this work.  While one reviewer was initially concerned about novelty and empirical scope, they changed their mind after sufficient back and forth with the authors, who provided additional experiments upon request.  Reviewers now all agree on both theoretical novelty, in the sense that this is an interesting new unifying framework, and sufficient empirical validation.